# The Application of the Modified Lindstedt–Poincaré Method to Solve the Nonlinear Vibration Problem of Exponentially Graded Laminated Plates on Elastic Foundations

Mahmure Avey [1,2,3], Francesco Tornabene [4], Nigar Mahar Aslanova [5] and Abdullah H. Sofiyev [6,7,8,*]

1  Department of Mathematical Engineering, Graduate School of Istanbul Technical University, Maslak 34469, Istanbul, Turkey; avey22@itu.edu.tr
2  Analytical Information Resources Center of UNEC, Azerbaijan State Economics University, Baku AZ1001, Azerbaijan
3  Application and Research Center, Istanbul Ticaret University, Beyoglu 34445, Istanbul, Turkey
4  Department of Innovation Engineering, University of Salento, I-73100 Lecce, Italy; francesco.tornabene@unibo.it
5  Department of Mathematics, Azerbaijan University of Architecture and Construction, Baku AZ1073, Azerbaijan; nigar.aslanova@azmiu.edu.az
6  Department of Mathematics, Istanbul Ticaret University, Beyoglu 34445, Istanbul, Turkey
7  Scientific Research Department, Azerbaijan University of Architecture and Construction, Baku AZ1073, Azerbaijan
8  Scientific Research Centers for Composition Materials of UNEC, Azerbaijan State Economic University, Baku AZ1001, Azerbaijan
*  Correspondence: aavey@ticaret.edu.tr

**Abstract:** The solution of the nonlinear (NL) vibration problem of the interaction of laminated plates made of exponentially graded orthotropic layers (EGOLs) with elastic foundations within the Kirchhoff–Love theory (KLT) is developed using the modified Lindstedt–Poincaré method for the first time. Young's modulus and the material density of the orthotropic layers of laminated plates are assumed to vary exponentially in the direction of thickness, and Poisson's ratio is assumed to be constant. The governing equations are derived as equations of motion and compatibility using the stress–strain relationship within the framework of KLT and von Karman-type nonlinear theory. NL partial differential equations are reduced to NL ordinary differential equations by the Galerkin method and solved by using the modified Lindstedt–Poincaré method to obtain unique amplitude-dependent expressions for the NL frequency. The proposed solution is validated by comparing the results for laminated plates consisting of exponentially graded orthotropic layers with the results for laminated homogeneous orthotropic plates. Finally, a series of examples are presented to illustrate numerical results on the nonlinear frequency of rectangular plates composed of homogeneous and exponentially graded layers. The effects of the exponential change in the material gradient in the layers, the arrangement and number of the layers, the elastic foundations, the plate aspect ratio and the nonlinearity of the frequency are investigated.

**Keywords:** composites; inhomogeneity; NL equations; NL vibration; plates; ground effect; NL frequencies

**MSC:** 74B20; 74E05; 74E10; 74E30; 74F10; 74H10; 74H45; 74K20

## 1. Introduction

Laminated composite plates continue to be widely used in various fields of transportation, construction, aerospace, nuclear and fossil energy, chemistry and petrochemicals. The application of anisotropic and functionally graded materials increases the bearing capacity of structures. At the same time, using laminated plates in engineering design to reduce weight, increase durability and improve dynamic properties also provides economic

advantages. For this reason, the use of laminated functionally graded orthotropic plates in aircraft, satellites, light alloy structures of automobiles and other engineering structures remains up to date. Additionally, depending on the application areas, laminated plates can be designed with different layer arrangements and numbers. Among the above factors affecting the dynamic properties of the plate structure, the functionally graded orthotropic material properties of the layers, as well as the layer arrangement and number, are also very important. All of these factors add additional complexity to the structural safety and reliability assessment. In the vibration analysis of plate structures, material and geometric properties should be taken into account as much as possible as a connection, which is one of the issues that make scientists think.

The natural evolution of plate theory, which is widely presented in the literature, has led to the formation and development of laminated plate theories [1]. Detailed information about the first studies on the laminated anisotropic plates and their development history is presented in monographs [2]. In this study, the main formulations of boundary value problems in laminated anisotropic plate theory were formulated, methods to solve them were developed, and practical applications were realized.

The use of structural elements consisting of layers with piecewise continuously changing material properties in different areas of contemporary technology necessitates the reconsideration of their nonlinear vibration calculations. One of the main reasons for this phenomenon lies in the exponentially graded anisotropic nature of the layers that form laminated structural elements. As is known, although efforts to create analytical models that reflect the reality of the mechanical properties of exponentially graded anisotropic materials are always on the agenda, the number of modeling studies is limited [3–5]. However, many studies have been conducted on linear [6–11] and nonlinear [12–14] vibrations of functionally or exponentially graded orthotropic single and laminated plates. Among these, Fazzolari and Carrera [6,7] studied vibration and thermo-mechanical buckling analysis of anisotropic multilayered composite and sandwich plates by using refined variable kinematic theories. Bacciocchi and Tarantino [8] analyzed the natural frequency behavior of functionally graded orthotropic cross-ply plates based on the finite element method. Zhang et al. [9] examined accurate linear free vibration solutions of orthotropic rectangular thin plates by straightforward finite integral transform method. Krysko et al. [10] presented the mathematical modeling of planar physically nonlinear inhomogeneous plates with rectangular cuts in the three-dimensional formulation, and the finite element method was employed to solve the problem. Hashemi [11] presented layer-wise solutions for variable stiffness laminated composite sandwich plates with curvilinear fibers using the *p*-version of the finite element method. Dat et al. [12] presented a vibration analysis of an auxetic laminated plate with magneto-electro-elastic face sheets subjected to blast loading. Singh et al. [13] presented the analytical solution of the viscoelastic free vibration problem of in-plane functionally graded orthotropic plates integrated with piezoelectric sensors. Simões et al. [14] studied the maximization of bending and membrane frequencies of vibration of variable stiffness laminated composite plates by a genetic algorithm. Ribeiro and Akhavan [15] developed a *p*-version finite element with hierarchic basis functions for the solution of nonlinear vibrations of variable stiffness laminated composite plates. Gupta and Pradyumna [16] studied geometrically nonlinear dynamic analysis of variable stiffness laminated composite and sandwich panels by formulating higher-order structural models and considering the effect of curvilinear fiber angles. Ribeiro et al. [17] investigated nonlinear forced vibrations of variable stiffness plates on elastic supports using Kirchhoff's assumptions with von Karman's strain–displacement relation. Alimoradzadeh et al. [18] presented the analysis of nonlinear axial–lateral combined vibration of functionally graded fiber-reinforced laminated composite beams under aero-thermal loads using multiple time scales.

Since multilayer heterogeneous plates are generally in contact with elastic environments during their operation, it has become a necessity to consider the effects of anisotropy, heterogeneity, arrangement and number of layers on their behavior, as well as the influ-

ences of elastic environments on the vibration characteristics. One of the most realistic elastic foundation models was proposed by Pasternak [19], and the simplest version of this model is known to be the Winkler model. There are some studies on linear and nonlinear vibrations of single-layer heterogeneous anisotropic structural elements in contact with elastic media. Among these studies, Tornabene [20] studied free vibrations of anisotropic doubly curved shells and panels of revolution with a free-form meridian resting on Winkler–Pasternak elastic foundations using the generalized differential quadrature (GDQ) method. Haciyev et al. [21] studied linear free bending vibration analysis of thin bidirectionally exponentially graded orthotropic rectangular plates resting on two-parameter elastic foundations. Zenkour and coauthors [22–25] studied bending and linear free vibration analyses of exponentially graded single-layer micro- and macro-plates in surrounding media using different theories and analytical methods. Sofiyev et al. [26] investigated the influences of two-parameter elastic foundations on the nonlinear free vibration of anisotropic shallow shell structures with variable parameters. Song et al. [27] proposed an analytical method for linear vibration analysis of arbitrarily shaped non-homogeneous orthotropic plates of variable thickness resting on the Winkler–Pasternak foundation. Melaibari et al. [28,29] examined the static response of 2D functionally graded porous plates resting on elastic foundations using midplane and neutral surfaces with movable constraints and differential quadrature method. Jena et al. [30,31] presented the free vibration of functionally graded beams and the stability of nanobeams resting on the Winkler–Pasternak elastic foundation using a novel numerical approach. Zaoui et al. [32] presented a mathematical approach for the mechanical behavior analysis of FGM plates on elastic foundations. Kumar et al. [33] investigated the linear vibration response of the exponential functionally graded material plate with variable thickness resting on the orthotropic Pasternak foundation. Tornabene et al. [34] applied general boundary conditions for the static analysis of anisotropic double-curved shells based on Winkler foundations using the GDQ method.

The literature review reveals that the nonlinear vibration problem of multilayer plates consisting of exponentially changing interaction of orthotropic layers with elastic foundations has not been studied sufficiently, and the problem has not yet been solved with the Lindstedt–Poincaré method. Among the analytical methods available for solving nonlinear ordinary differential equations, the Lindstedt–Poincaré method is one of the most effective methods. In this study, the nonlinear vibration behavior of plates consisting of exponentially graded orthotropic layers resting on elastic foundations is solved by the modified Lindstedt–Poincaré method. The Lindstedt–Poincaré method is an extension of perturbation methods used to find the periodic solution of NL-ODEs by eliminating secular terms.

The results of this theoretical study of the nonlinear vibration of laminated plates composed of EGOLs in elastic media have practical implications for optimizing structural design, material selection, and performance in aerospace, marine, automotive, and related fields. The knowledge gained can also contribute to nonlinear vibration control and structural health monitoring. In addition, the Lindstedt–Poincaré method used to solve this problem can be applied to other laminated structural systems.

This paper can be summarized as follows. The introduction is designed in accordance with the subject heading in Section 1. In Section 2, the mathematical modeling of mechanical properties of exponentially graded laminated composite plates and elastic foundations is presented. In Section 3, the nonlinear dynamic model of exponentially graded laminated composite plates is introduced. In Section 4, the nonlinear differential equations for laminated plates consisting of exponentially graded composite layers interaction with elastic foundations are solved using the modified Lindstedt–Poincaré method. In Section 5, after performing numerical simulations to verify the validity of the analytical expressions, numerical examples are presented to verify the specificity of NL vibrations of exponentially graded laminated composite plates. The analysis conducted in this study makes an innovative contribution to the existing literature by comprehensively assessing

the linear and nonlinear oscillation behavior of laminated plates consisting of EGOLs in the elastic medium. Finally, Section 6 provides conclusions.

## 2. Modeling of Laminated Plates Composed of EGOLs

The schematic diagram of a laminated plate resting on a two-parameter elastic foundation composed of perfectly bonded exponentially graded orthotropic layers with total thickness, $h$, and lengths $a$ and $b$ on the $x_1$ and $x_2$ axes, respectively, are shown in Figure 1. $N$ is the total number of layers, and each EGOL has a thickness of $\delta = N/h$. The Cartesian coordinate system $Ox_1x_2x_3$ is used in the reference plane of the laminated plate, where the $x_1$ axis is taken along the length of the plate, the $x_2$ axis is taken along the width direction, and the $x_3$ axis is perpendicular to the reference plane. The origin is located at the left corner of the laminated EG orthotropic plate, as shown in Figure 1. The $x_3 = 0$ reference plane is located at the interface of the layers at even values of $N$, while at odd values of $N$, the reference plane is located at the reference plane of the middle layer. It is assumed that the main elasticity axes of all layers are parallel to the coordinate axes in the reference plane. The displacements in the $x_1$, $x_2$ and $x_3$ directions are indicated by $u$, $v$ and $w$.

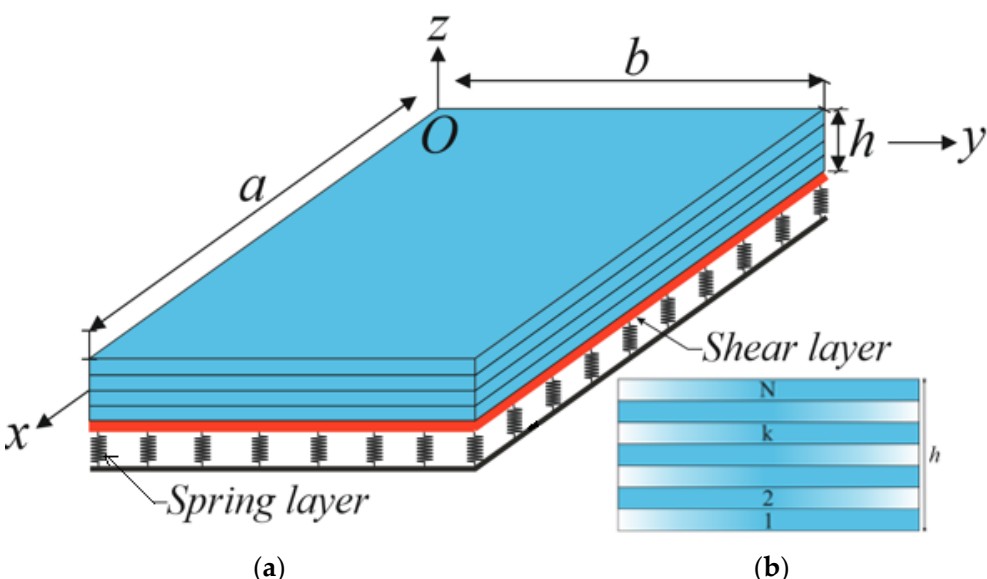

**(a)**             **(b)**

**Figure 1.** (**a**) Geometry and coordinate system of laminated exponentially graded plate on the two-parameter elastic foundation and (**b**) cross-section of laminated plate.

Assuming that the elastic foundation is well connected to the laminated plate in the large deflection region, the load–displacement relationship of the elastic foundation is modeled as [18]

$$R = K_w w - K_p \left( \frac{\partial^2 w}{\partial x_1^2} + \frac{\partial^2 w}{\partial x_2^2} \right) \tag{1}$$

where $R$ is the force per unit area, $K_w$ is the hardness coefficient of the Winkler foundation, and $K_p$ is the hardness coefficient of the shear layer of the elastic foundation.

Let $\Phi(x_1, x_2, t)$ be the Airy stress function for the stress resultants so that [1,2,35]

$$T_{11} = h \frac{\partial^2 \Phi}{\partial x_2^2}, \quad T_{22} = h \frac{\partial^2 \Phi}{\partial x_1^2}, \quad T_{12} = -h \frac{\partial^2 \Phi}{\partial x_1 \partial x_2} \tag{2}$$

In the following stages, $\Phi$ will be used instead of $\Phi(x_1, x_2, t)$.

The mechanical properties such as Young modulus $\left( E_{iiX_3}^{(k)} \right)$, shear modulus $\left( G_{12X_3}^{(k)} \right)$ and density $(\rho_{X_3}^{(k)})$ of lamina $k$th are exponential functions of the thickness coordinate and are defined as follows [3,26]:

$$E_{11X_3}^{(k)} = e^{\mu_1(X_3+1/2)} E_{11}^{0(k)}, \quad E_{22X_3}^{(k)} = e^{\mu_1(X_3+1/2)} E_{22}^{0(k)}, \quad G_{12X_3}^{(k)} = e^{\mu_1(X_3+1/2)} G_{12}^{0(k)}, \quad \rho_{X_3}^{(k)} = e^{\mu_2(X_3+1/2)} \rho^{0(k)}, \quad X_3 = x_3/h \quad (3)$$

where $\mu_i (i = 1, 2)$ are the material gradient parameters for the elasticity moduli and density, respectively, in the lamina $k$th, which varies in the range of $[-1, 1]$. Note that when $\mu_i = 0$, each layer of the plate is made of homogeneous orthotropic material.

## 3. Governing Equations

Based on the von Kármán-type nonlinear theory and the Kirchhoff–Love assumption, the motion and strain compatibility equations of exponentially graded orthotropic laminated plates, which includes the plate–foundation interaction, can be defined as [1,35]

$$\frac{\partial^2 M_{11}}{\partial x_1^2} + 2\frac{\partial^2 M_{12}}{\partial x_1 \partial x_2} + \frac{\partial^2 M_{22}}{\partial x_2^2} + T_{11}\frac{\partial^2 w}{\partial x_1^2} + 2T_{12}\frac{\partial^2 w}{\partial x_1 \partial x_2} + T_{22}\frac{\partial^2 w}{\partial x_2^2} - K_w w + K_p\left(\frac{\partial^2 w}{\partial x_1^2} + \frac{\partial^2 w}{\partial x_2^2}\right) = \rho_1 \frac{\partial^2 w}{\partial t^2} \quad (4)$$

$$\frac{\partial^2 e_{11}^0}{\partial x_2^2} + \frac{\partial^2 e_{22}^0}{\partial x_1^2} - \frac{\partial^2 \gamma_{12}^0}{\partial x_1 \partial x_2} = \left(\frac{\partial^2 w}{\partial x_1 \partial x_2}\right)^2 - \frac{\partial^2 w}{\partial x_1^2}\frac{\partial^2 w}{\partial x_2^2} \quad (5)$$

where $e_{ii}^0$ and $\gamma_{12}^0$ are strains in the reference plane, and the forces $(T_{ij})$, the moments $(M_{ij})$ and the density parameter $\rho_1$ are defined as follows [1,2,35]:

$$(T_{ij}, M_{ij}) = \sum_{k=1}^{N} \int_{(x_3)_{k-1}}^{(x_3)_k} \sigma_{ij}^{(k)}[1, x_3]\mathrm{d}x_3, \quad \rho_1 = \sum_{k=1}^{N} \int_{(x_3)_{k-1}}^{(x_3)_k} \rho_{X_3}^{(k)}\mathrm{d}x_3 \quad (i, j = 1, 2, 6), \quad (6)$$

in which $-\frac{h}{2} + \frac{(k-1)h}{N} \leq x_3 \leq -\frac{h}{2} + \frac{kh}{N}$.

According to the generalized Hooke's rule, the relationships between stresses $(\sigma_{ij}^{(k)})$ and strains of any point not located in the mid-plane ($e_{ii}$ and $\gamma_{12}$) for the layers contained in laminated plates consisting of EGOLs can be constructed as follows [3]:

$$\begin{pmatrix} \sigma_{11}^{(k)} \\ \sigma_{22}^{(k)} \\ \sigma_{12}^{(k)} \end{pmatrix} = \begin{bmatrix} \overline{E}_{11X_3}^{(k)} & \overline{E}_{12X_3}^{(k)} & 0 & 0 & 0 \\ \overline{E}_{21X_3}^{(k)} & \overline{E}_{22X_3}^{(k)} & 0 & 0 & 0 \\ 0 & 0 & \overline{E}_{66X_3}^{(k)} & 0 & 0 \end{bmatrix} \begin{bmatrix} e_{11} \\ e_{22} \\ \gamma_{12} \end{bmatrix} \quad (7)$$

where

$$\overline{E}_{11X_3}^{(k)} = \frac{E_{11X_3}^{(k)}}{1 - \nu_{12}^{(k)}\nu_{21}^{(k)}}, \quad \overline{E}_{12X_3}^{(k)} = \frac{\nu_{21}^{(k)} E_{11X_3}^{(k)}}{1 - \nu_{12}^{(k)}\nu_{21}^{(k)}} = \frac{\nu_{12}^{(k)} E_{22X_3}^{(k)}}{1 - \nu_{12}^{(k)}\nu_{21}^{(k)}} = \overline{E}_{21X_3}^{(k)}, \quad \overline{E}_{22X_3}^{(k)} = \frac{E_{22X_3}^{(k)}}{1 - \nu_{12}^{(k)}\nu_{21}^{(k)}}, \quad \overline{E}_{66X_3}^{(k)} = G_{12X_3}^{(k)} \quad (8)$$

in which $\nu_{ij}^{(k)}$ are Poisson ratios in the lamina $k$th and are considered constant and $\nu_{21}^{(k)} E_{11}^{0(k)} = \nu_{12}^{(k)} E_{22}^{0(k)}$.

When Kirchhoff–Love plate theory and nonlinearity are applied together, the relationships between the strains at any point not in the mid-plane and the strains in the mid-plane of laminated plates consisting of EGOLs can be modeled as follows [35]:

$$[e_{11}, e_{22}, \gamma_{12}] = \left[e_{11}^0 - x_3\frac{\partial^2 w}{\partial x_1^2}, \quad e_{22}^0 - x_3\frac{\partial^2 w}{\partial x_2^2}, \quad \gamma_{12}^0 - 2x_3\frac{\partial^2 w}{\partial x_1 \partial x_2}\right] \quad (9)$$

where

$$\left[e_{11}^0, e_{22}^0, \gamma_{12}^0\right] = \left[\frac{\partial u}{\partial x_1} + \frac{1}{2}\left(\frac{\partial w}{\partial x_1}\right)^2, \quad \frac{\partial v}{\partial x_2} + \frac{1}{2}\left(\frac{\partial w}{\partial x_2}\right)^2, \quad \frac{\partial v}{\partial x_1} + \frac{\partial u}{\partial x_2} + \frac{\partial w}{\partial x_1}\frac{\partial w}{\partial x_2}\right] \quad (10)$$

Equation (9) is substituted into (7), and the resulting expressions are substituted into the integrals of (6), and considering Expression (2), the strains in the middle plane and the expressions of force and moment components based on $\Phi$ and $w$ functions are derived. If those expressions are substituted into Equations (4) and (5), the nonlinear motion and deformation compatibility equations for laminated plates consisting of EGOLs, which includes the plate–foundation interaction within KLT, become as follows:

$$
\begin{aligned}
& h\left[c_{12}\frac{\partial^4\Phi}{\partial x_1^4} + (c_{11} - 2c_{31} + c_{22})\frac{\partial^4\Phi}{\partial x_1^2\partial x_2^2} + c_{21}\frac{\partial^4\Phi}{\partial x_2^4}\right] - c_{13}\frac{\partial^4 w}{\partial x_1^4} - (c_{14} + 2c_{32} + c_{23})\frac{\partial^4 w}{\partial x_1^2\partial x_2^2} \\
& -c_{24}\frac{\partial^4 w}{\partial x_2^4} + h\left(\frac{\partial^2\Phi}{\partial x_2^2}\frac{\partial^2 w}{\partial x_1^2} - 2\frac{\partial^2\Phi}{\partial x_1\partial x_2}\frac{\partial^2 w}{\partial x_1\partial x_2} + \frac{\partial^2\Phi}{\partial x_1^2}\frac{\partial^2 w}{\partial x_2^2}\right) - K_w w + K_p\left(\frac{\partial^2 w}{\partial x_1^2} + \frac{\partial^2 w}{\partial x_2^2}\right) - \rho_1\frac{\partial^2 w}{\partial t^2} = 0
\end{aligned}
\tag{11}
$$

$$
\begin{aligned}
& h\left[b_{11}\frac{\partial^4\Phi}{\partial x_2^4} + (b_{12} + b_{21} + b_{31})\frac{\partial^4\Phi}{\partial x_1^2\partial x_2^2} + b_{22}\frac{\partial^4\Phi}{\partial x_1^4}\right] - b_{23}\frac{\partial^4 w}{\partial x_1^4} \\
& - (b_{24} + b_{13} - b_{32})\frac{\partial^4 w}{\partial x_1^2\partial x_2^2} - b_{14}\frac{\partial^4 w}{\partial x_2^4} - \left(\frac{\partial^2 w}{\partial x_1\partial x_2}\right)^2 + \frac{\partial^2 w}{\partial x_1^2}\frac{\partial^2 w}{\partial x_2^2} = 0
\end{aligned}
\tag{12}
$$

where

$$
\begin{aligned}
& c_{11} = s_{11}^1 b_{11} + s_{12}^1 b_{21},\ \ c_{12} = s_{11}^1 b_{12} + s_{12}^1 b_{11},\ \ c_{13} = s_{11}^1 b_{13} + s_{12}^1 b_{23} + s_{11}^2,\ \ c_{14} = s_{11}^1 b_{14} + s_{12}^1 b_{24} + s_{12}^2, c_{21} = s_{21}^1 b_{11} + s_{22}^1 b_{21}, \\
& c_{22} = s_{21}^1 b_{12} + s_{22}^1 b_{22},\ \ c_{23} = s_{21}^1 b_{13} + s_{22}^1 b_{23} + s_{21}^2,\ \ c_{24} = s_{21}^1 b_{14} + s_{22}^1 b_{24} + s_{22}^2,\ \ c_{31} = s_{66}^1 b_{35},\ \ c_{32} = s_{66}^1 b_{32} + 2s_{66}^2, \\
& b_{11} = \frac{s_{22}^0}{s},\ \ b_{12} = -\frac{s_{12}^0}{s},\ \ b_{13} = \frac{s_{12}^0 s_{21}^1 - s_{11}^1 s_{22}^0}{s},\ \ b_{14} = \frac{s_{12}^0 s_{22}^1 - s_{12}^1 s_{22}^0}{s},\ \ b_{21} = -\frac{s_{21}^0}{s},\ \ b_{22} = \frac{s_{11}^0}{s},\ \ b_{23} = \frac{s_{11}^1 s_{21}^0 - s_{21}^1 s_{11}^0}{s}, \\
& b_{24} = \frac{s_{12}^1 s_{21}^0 - s_{22}^1 s_{11}^0}{s},\ \ b_{31} = \frac{1}{s_{66}^0},\ \ b_{32} = -\frac{2s_{66}^1}{s_{66}^0}, s = s_{11}^0 s_{22}^0 - s_{12}^0 s_{21}^0
\end{aligned}
\tag{13}
$$

in which

$$
s_{ii}^{n_1} = \sum_{k=1}^{N}\int_{z_{k-1}}^{z_k}\overline{E}_{iiX_3}^{(k)}x_3^{n_1}\mathrm{d}x_3,\ \ s_{ij}^{n_1} = \sum_{k=1}^{N}\int_{z_{k-1}}^{z_k}\nu_{ji}^{(k)}\overline{E}_{iiX_3}^{(k)}x_3^{n_1}\mathrm{d}x_3,\ \ (n_1 = 0,1,2;\ \ i = 1,2,6, j = 1,2)
\tag{14}
$$

## 4. Solution Method

The following simply supported boundary conditions are enjoined at the side edges of the laminated plate consisting of the EGOLs [1,35]:

$$
\begin{aligned}
& w = 0,\ \ \frac{\partial^2 w}{\partial x_2^2} = 0\ \text{if}\ x_1 = 0\ \text{and}\ \ a \\
& w = 0,\ \ \frac{\partial^2 w}{\partial x_1^2} = 0\ \text{if}\ x_2 = 0\ \text{and}\ \ b
\end{aligned}
\tag{15}
$$

Approximation functions that provide simply supported boundary conditions are sought as follows [35]:

$$
w = w_1(t)\sin(\lambda_1 x_1)\sin(\lambda_2 x_2),
\tag{16}
$$

where $w_1(t)$ represents a time-dependent function, and $\lambda_1 = \frac{m\pi}{a}$ and $\lambda_2 = \frac{n\pi}{b}$ are the wave parameters in which $(m, n)$ are mode numbers.

To find the Airy stress function from the particular solution of the non-homogeneous differential Equation (12), Relation (14) is substituted into the differential equation in question, and after some mathematical operations, the Airy stress function can be expressed with $w_1(t)$ as follows:

$$
\Phi = p_1 w_1^2(t)\cos(2\lambda_1 x_1) + p_2 w_1^2(t)\cos(2\lambda_2 x_2) + p_3 w_1(t)\sin(\lambda_1 x_1)\sin(\lambda_2 x_2)
\tag{17}
$$

The symbols in (17) are defined as follows:

$$
p_1 = \frac{1}{32b_{22}h}\left(\frac{\lambda_2}{\lambda_1}\right)^2,\ \ p_2 = \frac{1}{32b_{11}h}\left(\frac{\lambda_1}{\lambda_2}\right)^2,\ \ p_3 = \frac{b_{23}\lambda_1^4 + (b_{24} + b_{13} - b_{32})\lambda_1^2\lambda_2^2 + b_{14}\lambda_2^4}{h\left[b_{11}\lambda_2^4 + (b_{12} + b_{21} + b_{31})\lambda_1^2\lambda_2^2 + b_{22}\lambda_1^4\right]}
\tag{18}
$$

The Galerkin method is applied to the nonlinear equation of motion; that is, Equation (11), in the region $\{(x_1, x_2), \ 0 \leq x_1 \leq a, \ 0 \leq x_2 \leq b\}$, is as follows:

$$
\begin{aligned}
\int_0^a \int_0^b &\left[ -\rho_1 \frac{\partial^2 w}{\partial t^2} + c_{12} h \frac{\partial^4 \Phi}{\partial x_1^4} + (c_{11} - 2c_{31} + c_{22}) h \frac{\partial^4 \Phi}{\partial x_1^2 \partial x_2^2} + c_{21} h \frac{\partial^4 \Phi}{\partial x_2^4} - c_{13} \frac{\partial^4 w}{\partial x_1^4} - (c_{14} + 2c_{32} + c_{23}) \frac{\partial^4 w}{\partial x_1^2 \partial x_2^2} \right. \\
&\left. - c_{24} \frac{\partial^4 w}{\partial x_2^4} + h \left( \frac{\partial^2 \Phi}{\partial x_2^2} \frac{\partial^2 w}{\partial x_1^2} - 2 \frac{\partial^2 \Phi}{\partial x_1 \partial x_2} \frac{\partial^2 w}{\partial x_1 \partial x_2} + \frac{\partial^2 \Phi}{\partial x_1^2} \frac{\partial^2 w}{\partial x_2^2} \right) - K_w w + K_p \left( \frac{\partial^2 w}{\partial x_1^2} + \frac{\partial^2 w}{\partial x_2^2} \right) \right] \sin(\lambda_1 x_1) \sin(\lambda_2 x_2) dx_2 dx_1 = 0
\end{aligned}
\tag{19}
$$

After substituting Relations (16) and (17) into Equation (19) and performing the integration and taking into account $\overline{W} = w_1 / h$, the following nonlinear ordinary differential equation is obtained for the nonlinear free vibration of laminated plates made of EGOLs which includes the plate–foundation interaction:

$$
\frac{\mathrm{d}^2 \overline{W}(t)}{\mathrm{d}t^2} + k_{1wp} \overline{W}(t) + k_2 \overline{W}^2(t) + k_3 \overline{W}^3(t) = 0
\tag{20}
$$

where $k_{1wp} = \omega_{0wp}^2$, in which $\omega_{0wp}$ is the linear frequency of laminated plates made of EGOLs on the Pasternak elastic foundation and is defined as

$$
\omega_{0wp} = \sqrt{ \frac{1}{\rho_1} \begin{Bmatrix} c_{13} \lambda_1^4 + (c_{14} + 2c_{32} + c_{24}) \lambda_1^2 \lambda_2^2 + c_{24} \lambda_2^4 + K_w + K_p (\lambda_1^2 + \lambda_2^2) \\ - \left[ c_{12} \lambda_1^4 + (c_{11} - 2c_{31} + c_{22}) \lambda_1^2 \lambda_2^2 + c_{21} \lambda_2^4 \right] h p_3 \end{Bmatrix} }
\tag{21}
$$

Other symbols in (20) are

$$
k_2 = \frac{16 h^2}{3ab} \frac{1}{\rho_1} \left( \frac{4 c_{12} \lambda_1^3 p_1}{\lambda_2} + \frac{4 c_{21} \lambda_2^3 p_2}{\lambda_1} - \frac{\lambda_2 \lambda_1 p_3}{2} \right) \left[ 1 - (-1)^m - (-1)^n + (-1)^{m+n} \right], \quad k_3 = \frac{2 h^3 \lambda_1^2 \lambda_2^2 (p_1 + p_2)}{\rho_1}
\tag{22}
$$

The basic idea of the modified method is introducing a new variable [36–38]:

$$
t = \omega(\varepsilon) \tau
\tag{23}
$$

where $\omega$ is the frequency, and $\varepsilon$ is a small parameter. Equation (20) then becomes

$$
\frac{\mathrm{d}^2 \overline{W}}{\mathrm{d}\tau^2} + \alpha \overline{W} + \varepsilon \beta \overline{W}^2 + \varepsilon \gamma \overline{W}^3 = 0
\tag{24}
$$

where

$$
\alpha = k_{1wp} \omega^{-2}, \quad \beta = \frac{k_2 \omega^{-2}}{\varepsilon}, \quad \gamma = \frac{k_3 \omega^{-2}}{\varepsilon}
\tag{25}
$$

To solve Equation (24), $\overline{W}$ and $\alpha$ are expanded by powers of $\varepsilon$ as follows [36]:

$$
\begin{aligned}
\overline{W} &= \overline{W}_0 + \varepsilon \overline{W}_1 + \varepsilon^2 \overline{W}_2 + O(\varepsilon^3) \\
\alpha &= \Omega^2 + \varepsilon \omega_1 + \varepsilon^2 \omega_2 + O(\varepsilon^3)
\end{aligned}
\tag{26}
$$

where $\omega_i$ and $\overline{W}_i$ are the unknown parameters.

By substituting (26) into Equation (24) and then collecting the coefficients of the same powers of $\varepsilon^i (i = 0, 1, 2)$, we obtain

$$
\varepsilon^0 : \quad \frac{\mathrm{d}^2 \overline{W}_0}{\mathrm{d}\tau^2} + \Omega^2 \overline{W}_0 = 0
\tag{27}
$$

$$
\varepsilon^1 : \quad \frac{\mathrm{d}^2 \overline{W}_1}{\mathrm{d}\tau^2} + \Omega^2 \overline{W}_1 = -\omega_1 \overline{W}_0 - \beta \overline{W}_0^2 - \gamma \overline{W}_0^3
\tag{28}
$$

$$
\varepsilon^2 : \quad \frac{\mathrm{d}^2 \overline{W}_2}{\mathrm{d}\tau^2} + \Omega^2 \overline{W}_2 = -\omega_2 \overline{W}_0 - \omega_1 \overline{W}_1 - 2\beta \overline{W}_0 \overline{W}_1 - 3\gamma \overline{W}_1 \overline{W}_0^2
\tag{29}
$$

Solving Equation (27) with the initial conditions $\overline{W}_0 = f$, $\frac{d\overline{W}_0}{d\tau} = 0$, we obtain $\overline{W}_0 = f\cos(\Omega\tau)$, and substituting the expression for the $\overline{W}_0$ into (28) and after eliminating the secular terms, one obtains

$$\omega_1 = -\frac{3}{4}\gamma f^2 \tag{30}$$

Considering (30) and solving (28) with the initial conditions of $\overline{W}_1 = \frac{d\overline{W}_1}{d\tau} = 0$ yields

$$\overline{W}_1 = \frac{f^2}{\Omega^2}\left[\left(\frac{\beta}{3} - \frac{\gamma f}{32}\right)\cos(\Omega\tau) + \frac{\beta[\cos(2\Omega\tau) - 3]}{6} + \frac{\gamma f}{32}\cos(3\Omega\tau)\right] \tag{31}$$

Substituting the expression for $\overline{W}_0$ and $\overline{W}_1$ into Equation (29) and solving with the initial conditions of $\overline{W}_2 = \frac{d\overline{W}_2}{d\tau} = 0$, leading to the elimination of the secular term from the resulting equation, yields

$$\omega_2 = \frac{f^2}{2\Omega^2}\left(\frac{5\beta^2}{3} - \beta\gamma f + \frac{3\gamma^2 f^2}{64}\right) \tag{32}$$

When expressions for the $\omega_1$ and $\omega_2$ are substituted into the second equation of the set (26), the following algebraic equation for the nonlinear frequency is obtained:

$$\Omega^4 - 2\Lambda_1\Omega^2 + \Lambda_2 = 0 \tag{33}$$

where

$$\Lambda_1 = \frac{1}{2}\left(\omega_{0wp}^2 + \frac{3}{4}k_3 f^2\right), \quad \Lambda_2 = \frac{1}{2}\left(\frac{5k_2^2 f^2}{3} - k_2 k_3 f^3 + \frac{3k_3^2 f^4}{64}\right) \tag{34}$$

The solution of the previous fourth-order algebraic equation gives the expression for the nonlinear frequency of laminated plates consisting of EGOLs, which includes the plate–foundation interaction:

$$\Omega_{wp}^{NL} = \sqrt{\Lambda_1 + \sqrt{\Lambda_1^2 - \Lambda_2}} \tag{35}$$

The following expression is used for dimensionless values of nonlinear frequency:

$$\Omega_{1wp}^{NL} = \Omega_{wp}^{NL} h\sqrt{\frac{\rho_0^{(k)}}{E_{11}^{0(k)}}} \tag{36}$$

The ratio of the nonlinear frequency to the linear frequency is expressed as

$$\frac{\Omega_{wp}^{NL}}{\omega_0} = \sqrt{\frac{\Lambda_1 + \sqrt{\Lambda_1^2 - \Lambda_2}}{\omega_0^2}} \tag{37}$$

where $\omega_0$ denotes the linear frequency of laminated plates consisting of EGOLs without a ground and is defined as follows:

$$\omega_0 = \sqrt{\frac{c_{13}\lambda_1^4 + (c_{14} + 2c_{32} + c_{24})\lambda_1^2\lambda_2^2 + c_{24}\lambda_2^4 - \left[c_{12}\lambda_1^4 + (c_{11} - 2c_{31} + c_{22})\lambda_1^2\lambda_2^2 + c_{21}\lambda_2^4\right]hp_3}{\rho_1}} \tag{38}$$

## 5. Results and Discussion

### 5.1. Comparative Examples

To validate the present formulations and results, some comparison examples are presented in this subsection, as shown in Tables 1 and 2. In this section, the accuracy of the formulas is first confirmed by comparisons in the literature. Since it is seen that the smallest

values of the frequency parameter occur at $(m, n) = (1, 1)$ while making calculations in comparisons and numerical analyses, it is not included in the tables and figures.

**Table 1.** The nonlinear frequency vs. linear frequency ratios of thin square single-layer and laminated homogeneous plates without elastic ground with simply supported boundary conditions (SS4).

| | $\Omega^{NL}/\omega_0$ | | | |
|---|---|---|---|---|
| | **Isotropic Single-Layer Plate (0°)** | | **Laminated Orthotropic Plate with (0°/90°/0°/90°/0°)** | |
| $f$ | Present Study | Singha and Daripa [39] | Present Study | Singha and Daripa [39] |
| 0.4 | 1.02024 | 1.02049 | 1.04105 | 1.04169 |
| 0.8 | 1.07835 | 1.07959 | 1.15441 | 1.15823 |
| 1.2 | 1.16808 | 1.17249 | 1.32014 | 1.33111 |

**Table 2.** Comparison of $\overline{\omega}_{0wp}$ for single-layer square isotropic plates on the two-parameter elastic foundation.

| Methods | Wave Number | | |
|---|---|---|---|
| | $\omega_{0wp}^{(1,1)}$ | $\omega_{0wp}^{(1,2)}$ | $\omega_{0wp}^{(2,1)}$ |
| Wang et al. [40] | 3.3400 | 5.9287 | 5.9287 |
| Zhong et al. [41] | 3.3417 | 5.9289 | 5.9289 |
| Present study | 3.3406 | 5.9329 | 5.9329 |

**Example 1.** *The nonlinear frequency vs. linear frequency ratio* $(\Omega^{NL}/\omega_0)$ *of thin square single-layer homogenous isotropic and laminated homogeneous orthotropic cross-ply plates without elastic ground* $(a/b = 1; a/h = 100)$ *within KLT for different nondimensional parameters f are compared with the results obtained in the study of Singha and Daripa [39], which used the finite element method (see Table 1). In our calculations, the special case of Equation (37), that is, the case of* $\mu_i = 0$, $k_i = 0$ ($i = 1, 2$), *is considered. The material properties for the* $(0°/90°/0°/90°0°)$ *array plate are taken from Singha and Daripa [39] and are as follows:*

$$E_{11}^{0(k)}/E_{22}^{0(k)} = 40, \ G_{12}^{0(k)}/E_{22}^{0(k)} = G_{13}^{0(k)}/E_{22}^{0(k)} = 0.6, \ G_{23}^{0(k)}/E_{22}^{0(k)} = 0.5, \ \nu_{12}^{(k)} = 0.25, \ \rho_0^{(k)} = 1$$

*The isotropic material properties that make up the single-layer plate are as follows:*

$$E_{11}^{0(k)} = E_{22}^{0(k)} = E^{0(k)}, \ G_{12}^{0(k)} = G_{13}^{0(k)} = G_{23}^{0(k)} = E^{0(k)}/\left[2(1+\nu^{(k)})\right], \ \nu^{(k)} = 0.3, \ \rho_0^{(k)} = 1$$

*It is clearly seen in Table 1 that the numerical results obtained in this study for the* $\Omega^{NL}/\omega_0$ *ratio of both monolayer and laminated plates are in good agreement with the finite element results of Singha and Daripa [39].*

**Example 2.** *In the second comparison, the accuracy in terms of dimensionless linear frequency for single-layer square isotropic plates resting on the Pasternak elastic foundation for* $b = 100 \ h$, $a = b$, $K_w/D^{0(1)} = 500$ *and* $K_p/D^{0(1)} = 10$ *is ensured by comparing it with Wang et al. [40] and Zhong et al. [41]. In this example, relationship (21) is used for* $\mu_i = 0$, $E_{11}^{0(k)} = E_{22}^{0(k)} = E^{0(k)}$, *and the dimensionless linear frequency parameter on the Pasternak elastic foundation is taken into account as* $\overline{\omega}_{0wp} = \omega_{0wp}\left(\frac{b}{\pi}\right)^2\sqrt{\frac{\rho^{0(1)}h}{D^{0(1)}}}$, *with* $D^{0(1)} = \frac{E^{0(1)}h^3}{12\left[1-\left(\nu^{0(1)}\right)^2\right]}$, *as in Zhong et al. [41]. Table 2 reveals consistent agreement between our formulation and the findings presented in Wang et al. [40] and Zhong et al. [41], although based on different approaches.*

### 5.2. Nonlinear Analysis in Elastic Media

In this subsection, the influences of two-parameter soil, non-homogeneity, and the number and arrangement of the layers on the nonlinear frequency of laminated plates consisting of exponentially graded layers are examined using the Maple 14 software program. In numerical examples, the Expressions (36) and (37) for the nonlinear vibration frequency of laminated plates consisting of exponentially graded layers interaction with elastic foundations are used. The exponentially graded orthotropic material types, i.e., $EG_{(1,1)}$, $EG_{(1,0)}$ and $EG_{(-1,0)}$ correspond to $\mu_1 = \mu_2 = 1$, $\mu_1 = 1, \mu_2 = 0$ and $\mu_1 = -1, \mu_2 = 0$, respectively. Furthermore $\mu_1 = \mu_2 = 0$ corresponds to a homogeneous case (H). The mechanical properties of a homogeneous carbon fiber-reinforced polymer in the layers are considered [2]: $E_{11}^{0(k)} = 138,600$ MPa, $E_{22}^{0(k)} = 8270$ MPa, $G_{12}^{0(k)} = 4120$ MPa, $G_{13}^{0(k)} = G_{23}^{0(k)} = 0.6 \times E_{22}^{0(k)}$, $\nu_{12}^{(k)} = 0.26$ and $\rho_0^{(k)} = 1824$ kg/m$^3$. In the calculations, the coefficients of two-parameter soil are taken into account as $K_w = 1 \times 10^6$(N/m$^3$), $K_p = 3 \times 10^4$(N/m) and $5 \times 10^4$(N/m), whereas $(K_w, K_p) = (0,0)$ corresponds to the example without soil. The sequence and number of layers forming the laminated plate used in the analyses are presented in Figure 2.

**Figure 2.** Cross-sections of laminated plates with various layer sequences and numbers.

In Tables 3 and 4, the variation in nonlinear vibration of multilayer plates with different arrangements and numbers, whose layers consist of the homogeneous orthotropic material (HOM) and layers of the exponentially graded orthotropic material (EGOM), respectively, according to the distribution of $f$, in cases with and without ground, for different $a/b$ with $a/h = 75$ are presented. The NL frequency values increase when $f$ and $a/b$ ratios increase in cases with and without ground, in all alignments and layers, and in the EGOM profiles. In the HMO and EGOM cases, increasing $f$ supports the increase in the NL frequency but also reveals that the ground effect makes the NL frequency values significant enough to be considered. Since each of the $EG_{(1,1)}$, $EG_{(1,0)}$ and $EG_{(-1,0)}$ profiles has their own unique effects on the NL frequency, separate evaluation and interpretation must be made for each profile. It is seen that the NL frequency values of laminated plates consisting of $EG_{(-1.0)}$-profiled layers are higher than the values of the homogeneous case and are lower in other EGOM profiles.

The effect of exponentially graded profiles on the NL frequency changes when the layer layout changes. As the laminated plates are compared with the (0°) single-layer plate, the most significant alignment influences on the NL frequency in the case without ground are 77.8% for the (90°/0°/90°) array plate consisting of layers with the $EG_{(-1,0)}$ profile at $f = 0.25$ and $a/b = 1.5$, while the lowest effect (−0.19%) is observed in the (90°/0°/90°) and (0°/90°/0°) array plates consisting of EG(−1.0) layers and it occurs for $f = 1.5$ and $a/b = 1$. In the absence of ground effect, the influence of the alignment on the NL frequency of laminated plates starting with (0°/...) in all EGOM profiles increases as $f$ increment. In the grounded case, the strongest influence of the alignment on the NL frequency occurs in the (90°/0°/90°) array plate with $EG_{(-1.0)}$-profiled layers, with 65.61% for $f = 1.5$ and

$a/b$ = 1.5, while the lowest effect is ($-0.17\%$), which occurs in the ($0°/90°/0°$) and ($90°/0°/90°$) alignment plate with EG$_{(1.0)}$ and EG$_{(1,1)}$ profiles at $f$ = 0.25 and $a/b$ = 0.5. As can be seen, taking the ground into account reduces the influence of the layer arrangement on the NL frequency.

**Table 3.** Variation in NL frequency for single-layer and laminated plates consisting of HOM-profiled layers with and without elastic foundation against the $f$ with different $a/b$.

| $\Omega_1^{NL}\times 10^3$ for $(K_w,K_p)=(0,0)$ | | | | | | | | |
|---|---|---|---|---|---|---|---|---|
| | **a/b = 0.5** | | | **a/b = 1.0** | | | **a/b = 1.5** | |
| $f$ | (0°) | (0°/90°) | (0°/90°/0°) | (0°) | (0°/90°) | (0°/90°/0°) | (0°) | (0°/90°) | (0°/90°/0°) |
| 0.25 | 0.526 | 0.270 | 0.515 | 0.566 | 0.393 | 0.566 | 0.658 | 0.665 | 0.694 |
| 0.5 | 0.551 | 0.297 | 0.534 | 0.591 | 0.428 | 0.591 | 0.684 | 0.727 | 0.742 |
| 1 | 0.642 | 0.386 | 0.602 | 0.681 | 0.546 | 0.681 | 0.780 | 0.936 | 0.911 |
| 1.5 | 0.770 | 0.501 | 0.700 | 0.809 | 0.700 | 0.810 | 0.919 | 1.207 | 1.138 |
| $f$ | (90°/0°/90°) | (0°/90°/90°/0°) | (90°/0°/0°/90°) | (90°/0°/90°) | (0°/90°/90°/0°) | (90°/0°/0°/90°) | (90°/0°/90°) | (0°/90°/90°/0°) | (90°/0°/0°/90°) |
| 0.25 | 0.230 | 0.494 | 0.273 | 0.566 | 0.566 | 0.566 | 1.185 | 0.757 | 1.145 |
| 0.5 | 0.254 | 0.510 | 0.299 | 0.591 | 0.591 | 0.591 | 1.228 | 0.813 | 1.182 |
| 1 | 0.331 | 0.566 | 0.388 | 0.681 | 0.681 | 0.681 | 1.388 | 1.005 | 1.322 |
| 1.5 | 0.430 | 0.650 | 0.502 | 0.810 | 0.810 | 0.810 | 1.619 | 1.261 | 1.526 |
| $\Omega_{1wp}^{NL} \times 10^3$ for $(K_w,K_p) = (1\times10^6,\ 5\times10^4)$ | | | | | | | | |
| $f$ | (0°) | (0°/90°) | (0°/90°/0°) | (0°) | (0°/90°) | (0°/90°/0°) | (0°) | (0°/90°) | (0°/90°/0°) |
| 0.25 | 0.655 | 0.474 | 0.646 | 0.721 | 0.595 | 0.721 | 0.844 | 0.849 | 0.872 |
| 0.5 | 0.675 | 0.489 | 0.661 | 0.741 | 0.618 | 0.741 | 0.864 | 0.899 | 0.911 |
| 1 | 0.751 | 0.549 | 0.717 | 0.814 | 0.705 | 0.815 | 0.942 | 1.075 | 1.053 |
| 1.5 | 0.863 | 0.635 | 0.801 | 0.924 | 0.830 | 0.925 | 1.060 | 1.317 | 1.255 |
| $f$ | (90°/0°/90°) | (0°/90°/90°/0°) | (90°/0°/0°/90°) | (90°/0°/90°) | (0°/90°/90°/0°) | (90°/0°/0°/90°) | (90°/0°/90°) | (0°/90°/90°/0°) | (90°/0°/0°/90°) |
| 0.25 | 0.452 | 0.629 | 0.475 | 0.721 | 0.721 | 0.721 | 1.297 | 0.923 | 1.261 |
| 0.5 | 0.465 | 0.641 | 0.491 | 0.741 | 0.741 | 0.741 | 1.337 | 0.969 | 1.295 |
| 1 | 0.511 | 0.687 | 0.550 | 0.815 | 0.815 | 0.815 | 1.485 | 1.135 | 1.423 |
| 1.5 | 0.580 | 0.758 | 0.636 | 0.925 | 0.925 | 0.925 | 1.703 | 1.367 | 1.615 |

When the profile effect on NL frequency values of laminated plates consisting of layers with EGOM profiles is compared among themselves, the most significant influence occurs in the laminated plate with the EG$_{(-1.0)}$ profile. For example, at $a/b$ = 0.5 and $f$ = 1.5, the highest effects of the EG$_{(-1.0)}$ profile on the NL frequency occur with 27.02% and 40.92% for the ($0°/90°$) array plate with and without ground, respectively, while the highest effect occurs in the unconstrained ($0°/90°/0°$) array plate with 29.44% and in the ($90°/0°/0°/90°$) alignment plate on the elastic foundation with 26.57%, at $a/b$ = 1.5 and $f$ = 1.5. It is clearly seen that considering the ground weakens the effect of EG profiles on the NL frequency.

In laminated plates consisting of homogeneous orthotropic layers, the most significant ground effect on NL frequency values occurs with 51.3% in the ($0°/90°$)-aligned plate at $a/b$ = 0.5 and $f$ = 0.25, while the weakest effect occurs with 5.2% in the ($90°/0°/90°$)-aligned plate for $a/b$ = 1.5 and $f$ = 1.5. In laminated plates consisting of EGOM-profiled layers, the most significant ground effect on NL frequency values occurs in the ($90°/0°/90°$)-aligned plates consisting of EG$_{(1,1)}$ and EG$_{(1,0)}$ profiles with 139% for $a/b$ = 0.5 and $f$ = 0.25, while

the weakest influence occurs with 3.15% in the (90°/0°/90°) plates with EG$_{(-1,0)}$ profiles for $a/b = 1.5$ and $f = 1.5$.

**Table 4.** Variation in NL frequency for single-layer and laminated plates consisting of EGOM-profiled layers with and without elastic foundation against the $f$ with different $a/b$.

| $f$ | $\Omega_1^{NL} \times 10^3$ for $(K_w, K_p)=(0,0)$ | | | | | | | | | $\Omega_{1wp}^{NL} \times 10^3$ for $(K_w, K_p)=(1\times10^6, 5\times10^4)$ | | | | | | | | |
|---|---|---|---|---|---|---|---|---|---|---|---|---|---|---|---|---|---|---|
| | **(0°)** | | | **(0°/90°)** | | | **(0°/90°/0°)** | | | **(0°)** | | | **(0°/90°)** | | | **(0°/90°/0°)** | | |
| | EG (1,1) | EG (1,0) | EG (−1,0) | EG (1,1) | EG (1,0) | EG (−1,0) | EG (1,1) | EG (1,0) | EG (−1,0) | EG (1,1) | EG (1,0) | EG (−1,0) | EG (1,1) | EG (1,0) | EG (−1,0) | EG (1,1) | EG (1,0) | EG (−1,0) |
| | | | | | | | | | ***a/b = 0.5*** | | | | | | | | | |
| 0.25 | 0.514 | 0.408 | 0.673 | 0.260 | 0.207 | 0.361 | 0.496 | 0.394 | 0.651 | 0.710 | 0.564 | 0.778 | 0.554 | 0.441 | 0.531 | 0.697 | 0.554 | 0.759 |
| 0.5 | 0.539 | 0.429 | 0.707 | 0.283 | 0.225 | 0.402 | 0.515 | 0.410 | 0.676 | 0.729 | 0.579 | 0.808 | 0.565 | 0.449 | 0.560 | 0.711 | 0.565 | 0.780 |
| 1.0 | 0.632 | 0.502 | 0.829 | 0.359 | 0.285 | 0.537 | 0.586 | 0.466 | 0.769 | 0.800 | 0.636 | 0.916 | 0.607 | 0.483 | 0.663 | 0.764 | 0.607 | 0.863 |
| 1.5 | 0.762 | 0.606 | 0.999 | 0.459 | 0.365 | 0.706 | 0.688 | 0.547 | 0.903 | 0.906 | 0.720 | 1.072 | 0.671 | 0.533 | 0.806 | 0.845 | 0.672 | 0.984 |
| $f$ | **(90°/0°/90°)** | | | **(0°/90°/90°/0°)** | | | **(90°/0°/0°/90°)** | | | **(90°/0°/90°)** | | | **(0°/90°/90°/0°)** | | | **(90°/0°/0°/90°)** | | |
| 0.25 | 0.226 | 0.179 | 0.296 | 0.472 | 0.375 | 0.619 | 0.266 | 0.212 | 0.349 | 0.539 | 0.428 | 0.489 | 0.680 | 0.541 | 0.732 | 0.557 | 0.443 | 0.523 |
| 0.5 | 0.249 | 0.197 | 0.327 | 0.489 | 0.388 | 0.641 | 0.293 | 0.233 | 0.384 | 0.549 | 0.436 | 0.508 | 0.692 | 0.550 | 0.750 | 0.570 | 0.454 | 0.547 |
| 1.0 | 0.325 | 0.258 | 0.427 | 0.549 | 0.436 | 0.720 | 0.381 | 0.303 | 0.500 | 0.588 | 0.467 | 0.578 | 0.736 | 0.585 | 0.819 | 0.621 | 0.493 | 0.634 |
| 1.5 | 0.423 | 0.336 | 0.556 | 0.637 | 0.506 | 0.835 | 0.495 | 0.393 | 0.649 | 0.647 | 0.514 | 0.679 | 0.803 | 0.639 | 0.922 | 0.696 | 0.553 | 0.757 |
| $f$ | | | | | | | | | ***a/b = 1.0*** | | | | | | | | | |
| | **(0°)** | | | **(0°/90°)** | | | **(0°/90°/0°)** | | | **(0°)** | | | **(0°/90°)** | | | **(0°/90°/0°)** | | |
| 0.25 | 0.553 | 0.439 | 0.725 | 0.388 | 0.309 | 0.514 | 0.547 | 0.434 | 0.718 | 0.788 | 0.626 | 0.851 | 0.682 | 0.543 | 0.681 | 0.783 | 0.623 | 0.845 |
| 0.5 | 0.578 | 0.459 | 0.758 | 0.422 | 0.335 | 0.563 | 0.572 | 0.454 | 0.752 | 0.806 | 0.641 | 0.880 | 0.702 | 0.558 | 0.718 | 0.801 | 0.637 | 0.874 |
| 1 | 0.670 | 0.532 | 0.878 | 0.538 | 0.428 | 0.721 | 0.664 | 0.528 | 0.874 | 0.874 | 0.695 | 0.985 | 0.778 | 0.618 | 0.848 | 0.869 | 0.691 | 0.982 |
| 1.5 | 0.800 | 0.636 | 1.049 | 0.692 | 0.550 | 0.924 | 0.795 | 0.632 | 1.047 | 0.977 | 0.777 | 1.140 | 0.891 | 0.708 | 1.026 | 0.973 | 0.774 | 1.138 |
| $f$ | **(90°/0°/90°)** | | | **(0°/90°/90°/0°)** | | | **(90°/0°/0°/90°)** | | | **(90°/0°/90°)** | | | **(0°/90°/90°/0°)** | | | **(90°/0°/0°/90°)** | | |
| 0.25 | 0.547 | 0.434 | 0.718 | 0.544 | 0.433 | 0.714 | 0.544 | 0.433 | 0.714 | 0.783 | 0.623 | 0.845 | 0.782 | 0.622 | 0.842 | 0.782 | 0.622 | 0.842 |
| 0.5 | 0.572 | 0.454 | 0.752 | 0.570 | 0.453 | 0.747 | 0.570 | 0.453 | 0.747 | 0.801 | 0.637 | 0.874 | 0.800 | 0.636 | 0.871 | 0.800 | 0.636 | 0.871 |
| 1.0 | 0.664 | 0.528 | 0.874 | 0.663 | 0.527 | 0.870 | 0.663 | 0.527 | 0.870 | 0.869 | 0.691 | 0.982 | 0.869 | 0.691 | 0.978 | 0.869 | 0.691 | 0.978 |
| 1.5 | 0.795 | 0.632 | 1.047 | 0.794 | 0.632 | 1.042 | 0.794 | 0.632 | 1.042 | 0.973 | 0.774 | 1.138 | 0.973 | 0.773 | 1.134 | 0.973 | 0.773 | 1.134 |
| $f$ | | | | | | | | | ***a/b = 1.5*** | | | | | | | | | |
| | **(0°)** | | | **(0°/90°)** | | | **(0°/90°/0°)** | | | **(0°)** | | | **(0°/90°)** | | | **(0°/90°/0°)** | | |
| 0.25 | 0.643 | 0.511 | 0.842 | 0.670 | 0.532 | 0.854 | 0.674 | 0.536 | 0.886 | 0.924 | 0.734 | 0.994 | 0.943 | 0.749 | 1.004 | 0.946 | 0.752 | 1.031 |
| 0.5 | 0.669 | 0.532 | 0.878 | 0.738 | 0.587 | 0.928 | 0.723 | 0.575 | 0.951 | 0.942 | 0.749 | 1.024 | 0.992 | 0.789 | 1.068 | 0.981 | 0.780 | 1.088 |
| 1 | 0.767 | 0.610 | 1.006 | 0.968 | 0.769 | 1.176 | 0.892 | 0.709 | 1.174 | 1.014 | 0.807 | 1.136 | 1.173 | 0.933 | 1.289 | 1.111 | 0.884 | 1.288 |
| 1.5 | 0.908 | 0.721 | 1.190 | 1.263 | 1.004 | 1.499 | 1.118 | 0.889 | 1.473 | 1.124 | 0.894 | 1.302 | 1.427 | 1.134 | 1.589 | 1.300 | 1.034 | 1.565 |
| $f$ | **(90°/0°/90°)** | | | **(0°/90°/90°/0°)** | | | **(90°/0°/0°/90°)** | | | **(90°/0°/90°)** | | | **(0°/90°/90°/0°)** | | | **(90°/0°/0°/90°)** | | |
| 0.25 | 1.141 | 0.907 | 1.497 | 0.735 | 0.584 | 0.963 | 1.095 | 0.871 | 1.436 | 1.320 | 1.049 | 1.587 | 0.990 | 0.787 | 1.099 | 1.281 | 1.018 | 1.530 |
| 0.5 | 1.186 | 0.943 | 1.557 | 0.791 | 0.629 | 1.037 | 1.135 | 0.902 | 1.488 | 1.359 | 1.080 | 1.644 | 1.032 | 0.821 | 1.164 | 1.315 | 1.045 | 1.580 |
| 1.0 | 1.351 | 1.074 | 1.776 | 0.984 | 0.782 | 1.290 | 1.282 | 1.019 | 1.681 | 1.506 | 1.197 | 1.853 | 1.187 | 0.944 | 1.394 | 1.443 | 1.148 | 1.762 |
| 1.5 | 1.590 | 1.264 | 2.090 | 1.240 | 0.986 | 1.626 | 1.495 | 1.188 | 1.961 | 1.723 | 1.370 | 2.156 | 1.407 | 1.119 | 1.710 | 1.636 | 1.300 | 2.031 |

The distribution of the nonlinear frequency to linear frequency ratio (NLF/LF) for single-layer and laminated plates composed of H, $EG_{(1,1)}$, $EG_{(1,0)}$ and $EG_{(-1,0)}$ profiles with and without elastic ground against the $f$ with different numbers and sequences of layers are plotted in Tables 5 and 6 and in Figures 3–6 for different $a/h$ and when $a/b = 0.5$. Since the NLF/LF ratio is the same for $EG_{(1,1)}$ and $EG_{(1,0)}$ profiles, the same column is used in the tables. Although it is seen that the NLF/LF ratio increases when the $f$ increases in plates starting from $(0°/...)$ or $(90°/...)$ array layers, in cases with and without ground, that ratio increases more clearly in some aligned plates in the case with ground.

**Table 5.** Variation in NLF/LF frequency ratio for single-layer and laminated plates consisting of HOM-profiled layers with and without elastic foundation against the $f$ with different $a/h$.

| | $\Omega^{NL}/\omega_0$ for $(K_w,K_p)=(0,0)$ | | | | | | | | |
|---|---|---|---|---|---|---|---|---|---|
| | a/h = 50 | | | a/h = 75 | | | a/h = 100 | | |
| $f$ | (0°) | (0°/90°) | (0°/90°/0°) | (0°) | (0°/90°) | (0°/90°/0°) | (0°) | (0°/90°) | (0°/90°/0°) |
| 0.25 | 1.017 | 1.038 | 1.012 | 1.017 | 1.038 | 1.012 | 1.017 | 1.038 | 1.012 |
| 0.5 | 1.065 | 1.142 | 1.048 | 1.065 | 1.142 | 1.048 | 1.065 | 1.142 | 1.048 |
| 1 | 1.241 | 1.486 | 1.181 | 1.241 | 1.486 | 1.181 | 1.241 | 1.486 | 1.181 |
| 1.5 | 1.488 | 1.928 | 1.375 | 1.488 | 1.928 | 1.375 | 1.488 | 1.928 | 1.375 |
| $f$ | (90°/0°/90°) | (0°/90°/90°/0°) | (90°/0°/0°/90°) | (90°/0°/90°) | (0°/90°/90°/0°) | (90°/0°/0°/90°) | (90°/0°/90°) | (0°/90°/90°/0°) | (90°/0°/0°/90°) |
| 0.25 | 1.037 | 1.011 | 1.036 | 1.037 | 1.011 | 1.036 | 1.037 | 1.011 | 1.036 |
| 0.5 | 1.142 | 1.042 | 1.137 | 1.142 | 1.042 | 1.137 | 1.142 | 1.042 | 1.137 |
| 1 | 1.489 | 1.158 | 1.474 | 1.489 | 1.158 | 1.474 | 1.489 | 1.158 | 1.474 |
| 1.5 | 1.933 | 1.328 | 1.908 | 1.933 | 1.328 | 1.908 | 1.933 | 1.328 | 1.908 |
| | $\Omega_{wp}^{NL}/\omega_0$ for $(K_w, K_p) = (1 \times 10^6, 3 \times 10^4)$ | | | | | | | | |
| $f$ | (0°) | (0°/90°) | (0°/90°/0°) | (0°) | (0°/90°) | (0°/90°/0°) | (0°) | (0°/90°) | (0°/90°/0°) |
| 0.25 | 1.080 | 1.264 | 1.077 | 1.216 | 1.686 | 1.219 | 1.483 | 2.385 | 1.492 |
| 0.5 | 1.126 | 1.351 | 1.111 | 1.257 | 1.752 | 1.249 | 1.517 | 2.432 | 1.517 |
| 1 | 1.293 | 1.653 | 1.238 | 1.409 | 1.994 | 1.362 | 1.645 | 2.612 | 1.612 |
| 1.5 | 1.532 | 2.059 | 1.423 | 1.631 | 2.341 | 1.533 | 1.839 | 2.886 | 1.759 |
| $f$ | (90°/0°/90°) | (0°/90°/90°/0°) | (90°/0°/0°/90°) | (90°/0°/90°) | (0°/90°/90°/0°) | (90°/0°/0°/90°) | (90°/0°/90°) | (0°/90°/90°/0°) | (90°/0°/0°/90°) |
| 0.25 | 1.338 | 1.081 | 1.258 | 1.869 | 1.233 | 1.672 | 2.718 | 1.525 | 2.360 |
| 0.5 | 1.421 | 1.110 | 1.342 | 1.929 | 1.259 | 1.736 | 2.760 | 1.545 | 2.406 |
| 1 | 1.712 | 1.220 | 1.638 | 2.152 | 1.356 | 1.974 | 2.920 | 1.626 | 2.583 |
| 1.5 | 2.110 | 1.383 | 2.037 | 2.481 | 1.505 | 2.315 | 3.170 | 1.751 | 2.852 |

In the case without ground, the influences of EGOM profiles on the NLF/LF ratio are weak at small values of $f$, while those effects increase for all alignments with the subsequent increase in $f$ and show a more significant increase in plates with alignments starting with $(0°/...)$ and are independent of the $a/h$ ratio. In the presence of a ground effect, when the $a/h$ ratio increases for the selected $f$, the effect of EGOM profiles on the NLF/LF ratio increases significantly depending on the layer arrangement and number and shows a more significant increase in plates with alignments starting with $(90°/...)$. In the case of without ground, when $f$ increases for the selected $a/h$, the influence of EGOM profiles on the NLF/LF ratio rises, whereas in the presence of the ground, those effects decrease, although they are much more pronounced.

**Table 6.** Variation in NLF/LF ratio for single-layer and laminated plates consisting of EGOM-profiled layers with and without elastic foundation against the *f* with different $a/h$.

### $a/h = 50$

| | $\Omega^{NL}/\omega_0$ for $(K_w, K_p) = (0,0)$ | | | | | | $\Omega^{NL}_{wp}/\omega_0$ for $(K_w, K_p) = (1\times10^6, 3\times10^4)$ | | | | | |
| --- | --- | --- | --- | --- | --- | --- | --- | --- | --- | --- | --- | --- |
| | (0°) | | (0°/90°) | | (0°/90°/0°) | | (0°) | | (0°/90°) | | (0°/90°/0°) | |
| $f$ | $EG_{(1,1)}$, $EG_{(1,0)}$ | $EG_{(-1,0)}$ | $EG_{(1,1)}$, $EG_{(1,0)}$ | $EG_{(-1,0)}$ | $EG_{(1,1)}$, $EG_{(1,0)}$ | $EG_{(-1,0)}$ | $EG_{(1,1)}$, $EG_{(1,0)}$ | $EG_{(-1,0)}$ | $EG_{(1,1)}$, $EG_{(1,0)}$ | $EG_{(-1,0)}$ | $EG_{(1,1)}$, $EG_{(1,0)}$ | $EG_{(-1,0)}$ |
| 0.25 | 1.018 | 1.018 | 1.030 | 1.046 | 1.013 | 1.014 | 1.120 | 1.056 | 1.390 | 1.179 | 1.122 | 1.055 |
| 0.5 | 1.069 | 1.069 | 1.118 | 1.167 | 1.052 | 1.053 | 1.167 | 1.106 | 1.456 | 1.288 | 1.158 | 1.093 |
| 1 | 1.252 | 1.252 | 1.418 | 1.557 | 1.197 | 1.198 | 1.337 | 1.284 | 1.698 | 1.650 | 1.290 | 1.234 |
| 1.5 | 1.509 | 1.509 | 1.813 | 2.046 | 1.405 | 1.407 | 1.580 | 1.535 | 2.039 | 2.117 | 1.486 | 1.437 |
| $f$ | (90°/0°/90°) | | (0°/90°/90°/0°) | | (90°/0°/0°/90°) | | (90°/0°/90°) | | (0°/90°/90°/0°) | | (90°/0°/0°/90°) | |
| 0.25 | 1.037 | 1.039 | 1.012 | 1.012 | 1.037 | 1.037 | 1.501 | 1.230 | 1.131 | 1.057 | 1.386 | 1.177 |
| 0.5 | 1.142 | 1.146 | 1.047 | 1.047 | 1.140 | 1.140 | 1.575 | 1.321 | 1.162 | 1.091 | 1.465 | 1.269 |
| 1 | 1.493 | 1.498 | 1.175 | 1.176 | 1.484 | 1.483 | 1.845 | 1.636 | 1.280 | 1.215 | 1.745 | 1.585 |
| 1.5 | 1.941 | 1.947 | 1.363 | 1.364 | 1.925 | 1.924 | 2.224 | 2.055 | 1.454 | 1.398 | 2.133 | 2.003 |

### $a/h = 75$

| $f$ | (0°) | | (0°/90°) | | (0°/90°/0°) | | (0°) | | (0°/90°) | | (0°/90°/0°) | |
| --- | --- | --- | --- | --- | --- | --- | --- | --- | --- | --- | --- | --- |
| 0.25 | 1.018 | 1.018 | 1.030 | 1.046 | 1.013 | 1.014 | 1.333 | 1.144 | 2.002 | 1.448 | 1.347 | 1.148 |
| 0.5 | 1.069 | 1.069 | 1.118 | 1.167 | 1.052 | 1.053 | 1.372 | 1.189 | 2.049 | 1.538 | 1.377 | 1.183 |
| 1 | 1.252 | 1.252 | 1.418 | 1.557 | 1.197 | 1.198 | 1.519 | 1.357 | 2.227 | 1.851 | 1.490 | 1.314 |
| 1.5 | 1.509 | 1.509 | 1.813 | 2.046 | 1.405 | 1.407 | 1.737 | 1.597 | 2.497 | 2.278 | 1.662 | 1.507 |
| $f$ | (90°/0°/90°) | | (0°/90°/90°/0°) | | (90°/0°/0°/90°) | | (90°/0°/90°) | | (0°/90°/90°/0°) | | (90°/0°/0°/90°) | |
| 0.25 | 1.037 | 1.039 | 1.012 | 1.012 | 1.037 | 1.037 | 2.249 | 1.595 | 1.375 | 1.159 | 1.983 | 1.458 |
| 0.5 | 1.142 | 1.146 | 1.047 | 1.047 | 1.140 | 1.140 | 2.300 | 1.667 | 1.401 | 1.189 | 2.039 | 1.533 |
| 1 | 1.493 | 1.498 | 1.175 | 1.176 | 1.484 | 1.483 | 2.492 | 1.926 | 1.499 | 1.304 | 2.249 | 1.803 |
| 1.5 | 1.941 | 1.947 | 1.363 | 1.364 | 1.925 | 1.924 | 2.784 | 2.292 | 1.651 | 1.476 | 2.561 | 2.180 |

### $a/h = 100$

| $f$ | (0°) | | (0°/90°) | | (0°/90°/0°) | | (0°) | | (0°/90°) | | (0°/90°/0°) | |
| --- | --- | --- | --- | --- | --- | --- | --- | --- | --- | --- | --- | --- |
| 0.25 | 1.018 | 1.018 | 1.030 | 1.046 | 1.013 | 1.014 | 1.724 | 1.322 | 2.960 | 1.927 | 1.756 | 1.336 |
| 0.5 | 1.069 | 1.069 | 1.118 | 1.167 | 1.052 | 1.053 | 1.754 | 1.362 | 2.991 | 1.996 | 1.779 | 1.366 |
| 1 | 1.252 | 1.252 | 1.418 | 1.557 | 1.197 | 1.198 | 1.872 | 1.510 | 3.116 | 2.246 | 1.868 | 1.481 |
| 1.5 | 1.509 | 1.509 | 1.813 | 2.046 | 1.405 | 1.407 | 2.053 | 1.729 | 3.314 | 2.609 | 2.008 | 1.654 |
| $f$ | (90°/0°/90°) | | (0°/90°/90°/0°) | | (90°/0°/0°/90°) | | (90°/0°/90°) | | (0°/90°/90°/0°) | | (90°/0°/0°/90°) | |
| 0.25 | 1.037 | 1.039 | 1.012 | 1.012 | 1.037 | 1.037 | 3.388 | 2.215 | 1.813 | 1.363 | 2.922 | 1.955 |
| 0.5 | 1.142 | 1.146 | 1.047 | 1.047 | 1.140 | 1.140 | 3.422 | 2.267 | 1.833 | 1.389 | 2.960 | 2.011 |
| 1 | 1.493 | 1.498 | 1.175 | 1.176 | 1.484 | 1.483 | 3.554 | 2.464 | 1.909 | 1.488 | 3.109 | 2.224 |
| 1.5 | 1.941 | 1.947 | 1.363 | 1.364 | 1.925 | 1.924 | 3.764 | 2.760 | 2.030 | 1.641 | 3.342 | 2.539 |

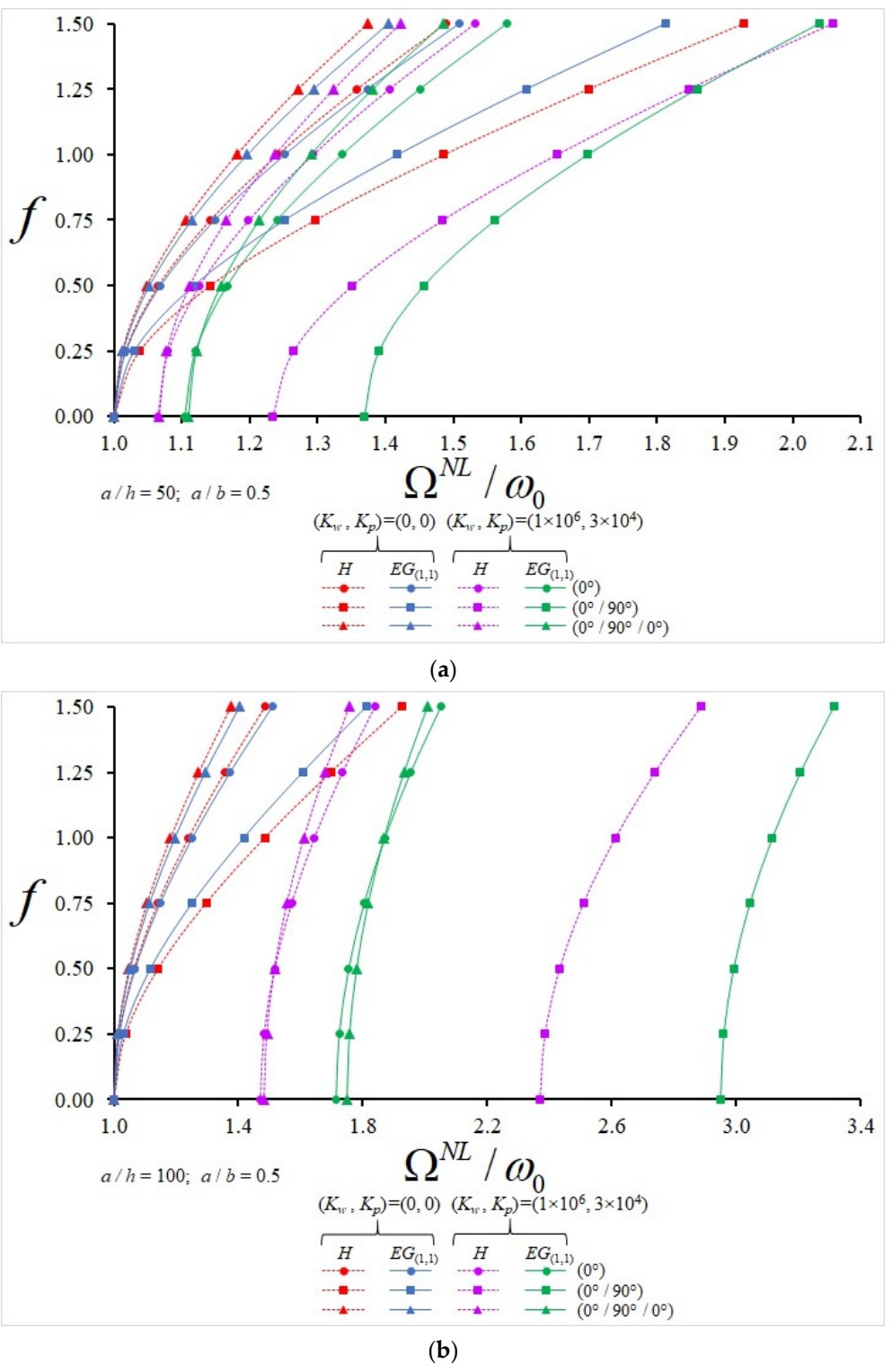

**Figure 3.** Variation in NLF/LF for single-layer and (0°)-, (0°/90°)- and (0°/90°/0°)-aligned plates consisting of HOM- and EG$_{(1,1)}$-profiled layers with and without elastic foundation against the *f* with (**a**) *a/h* = 50 and (**b**) *a/h* = 100.

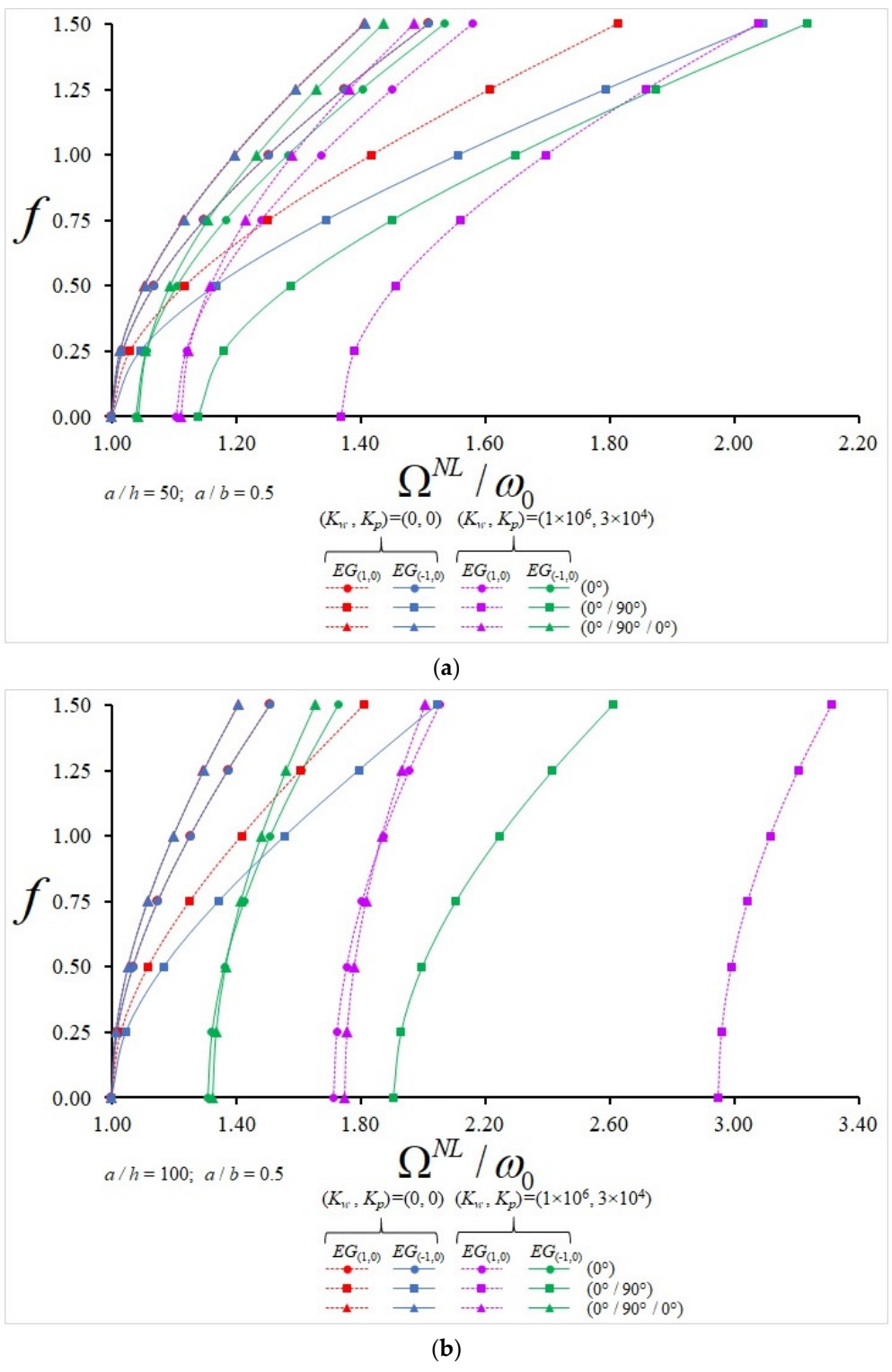

**Figure 4.** Variation in NLF/LF for single-layer and $(0°)$-, $(0°/90°)$- and $(0°/90°/0°)$-aligned plates consisting of $EG_{(1,0)}$- and $EG_{(-1,0)}$-profiled layers with and without elastic foundation against the $f$ with (**a**) $a/h = 50$ and (**b**) $a/h = 100$.

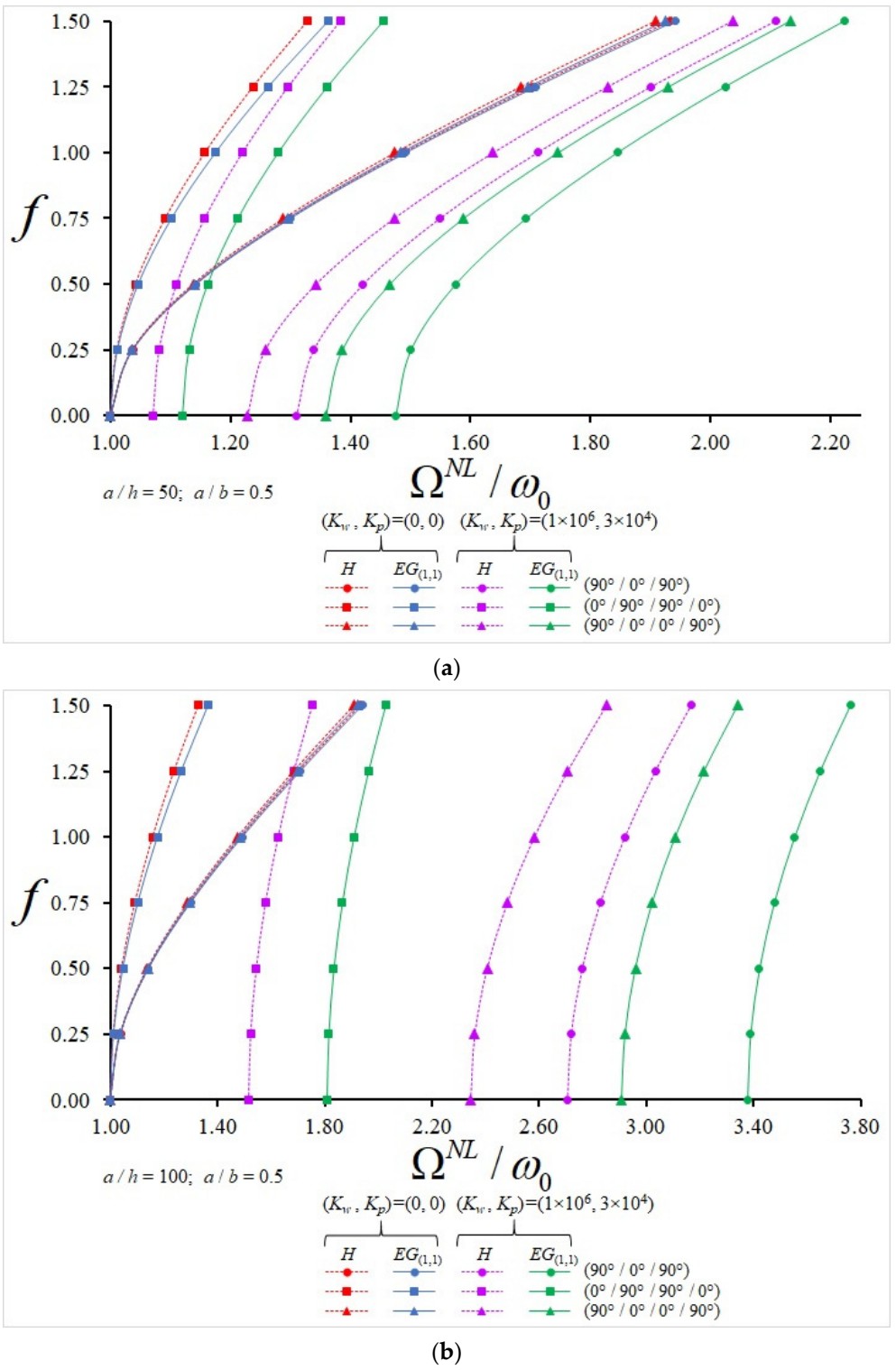

**Figure 5.** Variation in NLF/LF for single-layer and $(90°/0°/90°)$-, $(0°/90°/90°/0°)$- and $(90°/0°/0°/90°)$-aligned plates consisting of HOM- and $EG_{(1,1)}$-profiled layers with and without elastic foundation against the $f$ with (**a**) $a/h = 50$ and (**b**) $a/h = 100$.

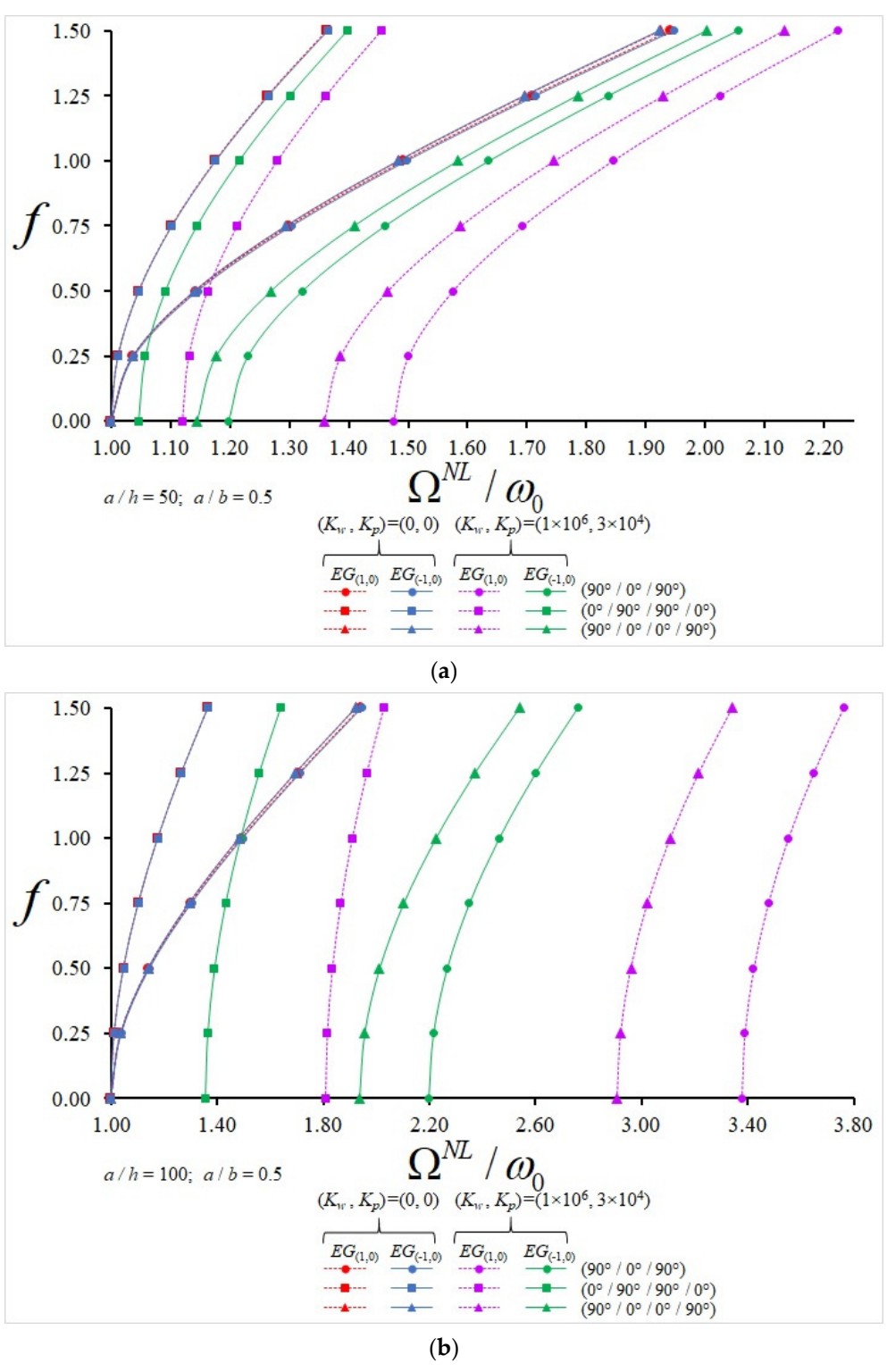

**Figure 6.** Variation in NLF/LF for single-layer and (90°/0°/90°)-, (0°/90°/90°/0°)- and (90°/0°/0°/90°)-aligned plates consisting of $EG_{(1,0)}$- and $EG_{(-1,0)}$-profiled layers with and without elastic foundation against the $f$ with (**a**) $a/h = 50$ and (**b**) $a/h = 100$.

For instance, in the ground case,

(a)  At $a/h = 50$ and $f = 0.25$, the influences of $EG_{(1,0)}$ and $EG_{(-1,0)}$ profiles on the NLF/LF ratio for plates with (0°/90°) alignment are 9.92% and (−6.75%); in the plates arranged in (0°/90°/0°), those effects are 4.17% and (−2.07%); in the plates arranged in

$(0°/90°/90°/0°)$, those influences are 4.64% and $(−2.18\%)$; in the plates arranged in $(90°/0°/90°)$, those influences are 12.17% and $(−8.1\%)$; and in the plates arranged in $(90°/0°/0°/90°)$, those effects are 10.15% and $(−6.4\%)$, respectively.

(b) At $a/h = 50$ and $f = 1.5$, the influences of $EG_{(1.0)}$ and $EG_{(−1.0)}$ profiles on the NLF/LF ratio for plates with $(0°/90°)$ alignment are $(−0.96\%)$ and 2.86%; in the plates arranged in $(0°/90°/0°)$, those influences are 4.37% and 0.97%; in the plates arranged in $(0°/90°/90°/0°)$, those influences are 5.17% and 1.11%; in the plates arranged in $(90°/0°/90°)$, those effects are 5.4% and $(−2.6\%)$; and in the plates arranged in $(90°/0°/0°/90°)$, those effects are 4.71% and $(−1.65\%)$, respectively.

(c) At $a/h = 100$ and $f = 0.25$, the influences of $EG_{(1.0)}$ and $EG_{(−1.0)}$ profiles on the NLF/LF ratio for plates with $(0°/90°)$ alignment are 24.1% and $(−19.2\%)$; in the plates arranged in $(0°/90°/0°)$, those influences are 17.68% and $(−10.5\%)$; in the plates arranged in $(0°/90°/90°/0°)$, those effects are 18.9% and $(−10.6\%)$; in the plates arranged in $(90°/0°/90°)$, those influences are 24.65% and $(−18.51\%)$; and in the plates arranged in $(90°/0°/0°/90°)$, those effects are 23.8% and $(−17.2\%)$, respectively.

(d) At $a/h = 100$ and $f = 1.5$, the influences of $EG_{(1.0)}$ and $EG_{(−1.0)}$ profiles on the NLF/LF ratio for plates with $(0°/90°)$ alignment are 14.85% and $(−9.58\%)$; in the plates arranged in $(0°/90°/0°)$, those effects are 14.2% and $(−5.93\%)$; in the plates arranged in $(0°/90°/90°/0°)$, those effects are 15.9% and $(−6.3\%)$; in the plates arranged in $(90°/0°/90°)$, those effects are 18.75% and $(−12.94\%)$; and in the plates arranged in $(90°/0°/0°/90°)$, those influences are 17.15% and $(−11\%)$, respectively (see Figures 3–6).

When the NLF/LF ratio of EGOM-profiled plates with different alignments consisting of two, three and four layers on the ground is compared with the single-layer plate $(0°)$, the alignment influences become evident with the increase in $f$ at $a/h = 50$. While at $a/h = 100$, as $f$ increases, the alignment effect reduces for the $EG_{(1,0)}$ profile, and that influence changes irregularly in the $EG_{(−1,0)}$ profile. Among all the arrays, the most significant influence on the NLF/LF ratio occurs in the $(90°/0°/90°)$ array plate with EG(1,0) profile with 96.5% and 83.4% at $f = 0.25$ and $f = 1.5$, respectively, while those effects for the $EG_{(−1,0)}$ profile are 67.5% and 59.64% when $f = 0.25$ and $f = 1.5$, respectively, as $a/h = 100$ (also see Figures 3 and 5).

In comparing the $(0°)$-monolayer plate with the layered plates in the case of without ground, the most significant alignment effect on the NLF/LF ratio occurs in the $(0°/90°)$-aligned plate and is approximately 35.61% at $f = 1.5$ for the $EG_{(−1,0)}$ profile and is independent of the $a/h$ ratio. As the $f$ ratio increases from 0.25 to 1.5 for all $a/h$, the alignment effect increases up to 29% in three-layer plates starting with $(90°/…)$, while in aligned plates starting with $(0°/…)$, that influence increases up to $(−6.9\%)$. In four-layer plates starting with $(90°/…)$, it is observed that the alignment effect increases up to 27.55%, whereas starting with $(0°/…)$-aligned plates, it increases up to $(−9.6\%)$ as $f$ rises from 0.25 to 1.5 for all $a/h$. In all layered plates, the effect of the soil on the NLF/LF ratio decreases significantly as $f$ increases, while the increase in the $a/h$ increases that effect significantly. The most significant ground effect on the NLF/LF ratio of all laminated plates occurs in the $(/90°/0°/90°)$-aligned plates for $EG_{(1,0)}$ and $EG_{(1,1)}$ profiles at $a/h = 100$ and $f = 0.25$, while for the $EG_{(−1,0)}$ profile, that effect would be about twice as low.

## 6. Conclusions

This study is devoted to solving the problem of the NL vibration of laminated plates consisting of EGOLs interacting with a two-parameter elastic foundation using the Lindstedt–Poincaré method. The NL fundamental relations and basic equations of laminated plates with EGOM profiles in an elastic medium were derived in this study. The amplitude-dependent expressions for the NL frequency are obtained using Galerkin and the modified Lindstedt–Poincaré method. The accuracy of the proposed solution is confirmed by comparing it with reliable results in the literature. The current analytical solution may be a valid option for comparison with finite element simulations. In addition,

lamination schemes within the scope of two-dimensional theories in continuous environments enable fast and accurate solutions to the nonlinear vibration problem of cross-ply plates made of orthotropic materials. Therefore, compared with reliable numerical models, it can be applied to lattice and honeycomb plates without significant computational effort, as well as to plates composed of heterogeneous composite materials. The numerical results are interpreted and generalized as follows:

(a)  The NL frequency values increase when $f$ and $a/b$ ratios increase in cases with and without ground, in all alignments and layers and in exponentially graded orthotropic material profiles;

(b)  In the homogeneous orthotropic and exponentially graded orthotropic material cases, increasing $f$ supports the increase in the NL frequency but also reveals that the ground effect makes the NL frequency values significant enough to be considered;

(c)  Since each of the $EG_{(1,1)}$, $EG_{(1,0)}$ and $EG_{(-1,0)}$ profiles has their own unique effects on the NL frequency, separate evaluation and interpretation must be made for each profile.

(d)  NL frequency values of laminated plates consisting of $EG_{(-1.0)}$ profile layers are higher than the values of the homogeneous case and are lower in other exponentially graded orthotropic material profiles;

(e)  The effect of exponential graduation profiles on the NL frequency changes when the layer layout or number of layers changes;

(f)  Taking the ground into account reduces the influence of the layer arrangement on the NL frequency;

(g)  As the model effect on NL frequency values of laminated plate consisting of layers with exponentially graded orthotropic material profiles is compared among themselves, the most significant effect occurs in the plate with the $EG_{(-1.0)}$ profile;

(h)  The ground weakens the influence of EG profiles and layer arrangement on the NL frequency;

(i)  Although the NLF/LF ratio increases when the $f$ increases in plates starting from $(0°/\ldots)$ or $(90°/\ldots)$ array layers, in cases with and without ground, that ratio increases more clearly in some aligned plates in the case with a ground;

(j)  In the case without a ground, the influences of exponentially graded orthotropic material profiles on the NLF/LF ratio are weak at small values of $f$, while those effects increase for all alignments with the subsequent increase in $f$ and show a more significant increase in plates with alignments starting with $(0°/\ldots)$, which are independent of the $a/h$ ratio;

(k)  In the presence of ground effect, when the $a/h$ ratio increases for the selected f, the effect of exponentially graded orthotropic material profiles on the NLF/LF ratio increases significantly depending on the layer arrangement and number and shows a more significant increase in plates with alignments starting with $(90°/\ldots)$;

(l)  In the case of without the ground, when $f$ increases for the selected $a/h$, the influence of exponentially graded orthotropic material profiles on the NLF/LF ratio increases, whereas in the presence of the ground, those effects decrease, although they are much more pronounced;

(m)  When the NLF/LF ratio of exponentially graded orthotropic material-profiled plates with different alignments consisting of two, three and four layers on the ground is compared with the single-layer plate $(0°)$, the alignment effects become evident with the increase in $f$;

(n)  In all layered plates, the influence of the soil on the NLF/LF ratio decreases significantly as $f$ increases, while the increase in the $a/h$ increases that effect significantly.

**Author Contributions:** Conceptualization, M.A.; methodology, M.A.; software, M.A.; validation, M.A. and F.T.; formal analysis, M.A.; investigation, M.A.; resources, M.A. and N.M.A.; writing—original draft preparation, M.A.; writing—review and editing, M.A., F.T., N.M.A. and A.H.S. All authors have read and agreed to the published version of the manuscript.

**Funding:** This research received no external funding.

**Data Availability Statement:** No data were reported in this study.

**Conflicts of Interest:** The authors declare no conflicts of interest.

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
