# Peer review of "The Application of the Modified Lindstedt–Poincaré Method to Solve the Nonlinear Vibration Problem of Exponentially Graded Laminated Plates on Elastic Foundations"

_mathematics, doi:10.3390/math12050749_

Round 1
Reviewer 1 Report
Comments and Suggestions for Authors
The authors have provided the solution of the nonlinear (NL) vibration problem of laminated plates made of exponentially graded orthotropic layers (EGOLs) interaction with elastic foundations within Kirchhoff Love Theory (KLT) is developed using the modified Lindstedt-Poincaré method.
The work is good. It can be published in the present form. However, the author should provide the advantage of the present method of solution with Ref 38.
Comments on the Quality of English LanguageOk
Author Response
25.02.2024
EXPLANATION TO REVIEWER 1:
First of all, we would like to thank the highly respected Reviewer 1 for his/her comments and the time spent for them.
Reviewer 1
(x) I would not like to sign my review report
( ) I would like to sign my review report
Quality of English Language
( ) I am not qualified to assess the quality of English in this paper
( ) English very difficult to understand/incomprehensible
( ) Extensive editing of English language required
( ) Moderate editing of English language required
(x) Minor editing of English language required
( ) English language fine. No issues detected
| 
 Yes  | 
 Can be improved  | 
 Must be improved  | 
 Not applicable  | 
|
| 
 Does the introduction provide sufficient background and include all relevant references?  | 
 (x)  | 
 ( )  | 
 ( )  | 
 ( )  | 
| 
 Are all the cited references relevant to the research?  | 
 (x)  | 
 ( )  | 
 ( )  | 
 ( )  | 
| 
 Is the research design appropriate?  | 
 (x)  | 
 ( )  | 
 ( )  | 
 ( )  | 
| 
 Are the methods adequately described?  | 
 (x)  | 
 ( )  | 
 ( )  | 
 ( )  | 
| 
 Are the results clearly presented?  | 
 (x)  | 
 ( )  | 
 ( )  | 
 ( )  | 
| 
 Are the conclusions supported by the results?  | 
 (x)  | 
 ( )  | 
 ( )  | 
 ( )  | 
Comments and Suggestions for Authors
The authors have provided the solution of the nonlinear (NL) vibration problem of laminated plates made of exponentially graded orthotropic layers (EGOLs) interaction with elastic foundations within Kirchhoff Love Theory (KLT) is developed using the modified Lindstedt-Poincaré method.
The work is good. It can be published in the present form. However, the author should provide the advantage of the present method of solution with Ref 38.
EXPLANATION 1: Thanks, for your constructive and positive response. In Ref. 39, the problem was solved by the finite element method and homogeneous plates were taken into account. The most important advantage of this study compared to Ref. 39 is that the results obtained are valid for heterogeneous and homogeneous composite plates and the analytical solution is obtained.
Thank you for your constructive and positive report.

Reviewer 2 Report
Comments and Suggestions for Authors
Present manuscript entitled " Application of Modified Lindstedt-Poincaré Method to Solution of Non-Linear Vibration Problem of Laminated Exponentially Graded Plates on Elastic Foundations " provides new method in solving nonlinear vibration problems. The mathematical framework behind the Lindstedt-Poincaré method has been presented properly. The manuscript can be accepted after the following minor corrections are made.
1. Does the assumption considered have any impact on the accuracy of results? If so, please specify?
2. In page no.3, you can provide Young’s modulus instead of Young moduli.
3. If some practical engineering applications can be mentioned in the manuscript based on the implication of the current method.
4. If the dpi of the figures can be improved for further clarity.
5. Can the current modified Lindstedt-Poincaré method be used to analyze other types of composite structures?

Author Response
25.02.2024
EXPLANATION TO REVIEWER 2:
First of all, we would like to thank the highly respected Reviewer 2 for his/her comments and the time spent for them.
Reviewer 2
(x) I would not like to sign my review report
( ) I would like to sign my review report
Quality of English Language
( ) I am not qualified to assess the quality of English in this paper
( ) English very difficult to understand/incomprehensible
( ) Extensive editing of English language required
( ) Moderate editing of English language required
( ) Minor editing of English language required
(x) English language fine. No issues detected
| 
 Yes  | 
 Can be improved  | 
 Must be improved  | 
 Not applicable  | 
|
| 
 Does the introduction provide sufficient background and include all relevant references?  | 
 (x)  | 
 ( )  | 
 ( )  | 
 ( )  | 
| 
 Are all the cited references relevant to the research?  | 
 (x)  | 
 ( )  | 
 ( )  | 
 ( )  | 
| 
 Is the research design appropriate?  | 
 (x)  | 
 ( )  | 
 ( )  | 
 ( )  | 
| 
 Are the methods adequately described?  | 
 (x)  | 
 ( )  | 
 ( )  | 
 ( )  | 
| 
 Are the results clearly presented?  | 
 ( )  | 
 (x)  | 
 ( )  | 
 ( )  | 
| 
 Are the conclusions supported by the results?  | 
 ( )  | 
 (x)  | 
 ( )  | 
 ( )  | 
Comments and Suggestions for Authors
2 Reviewer Comments
Present manuscript entitled " Application of modified Lindstedt-Poincaré method to solution of non-linear vibration problem of laminated exponentially graded plates on elastic foundations " provides new method in solving nonlinear vibration problems. The mathematical framework behind the Lindstedt-Poincaré method has been presented properly. The manuscript can be accepted after the following minor corrections are made.
SUGGESTION 1: Does the assumption considered have any impact on the accuracy of results? If so, please specify?
EXPLANATION 1: Thanks. The assumption taken into account has the effect of ensuring the accuracy of the results. It shows that the results obtained within the framework of the expanded theory with these assumptions are in harmony with the finite element and other numerical results.
SUGGESTION 2: In page no.3, you can provide Young’s modulus instead of Young moduli.
EXPLANATION 2: Thanks. You are right. This typo is corrected in the revised manuscript.
SUGGESTION 3: If some practical engineering applications can be mentioned in the manuscript based on the implication of the current method.
EXPLANATION 3: Thanks. The following sentences are added to the revised manuscript.
The results of this theoretical study of the nonlinear vibration of laminated plates composed of EGOLs in elastic media have practical implications for optimizing structural design, material selection, and performance in aerospace, marine, automotive, and related fields. The knowledge gained can also contribute to nonlinear vibration control and structural health monitoring. In addition, the Lindstedt-Poincaré method used to solve this problem can be applied to other laminated structural systems.
SUGGESTION 4: If the dpi of the figures can be improved for further clarity.
EXPLANATION 4: Thanks. Improved dpi of figures as much as possible.
a)
b)
Figure 3. Variation of NLF/LF for single layer and (0°), (0°/90°) and (0°/90°/0°) aligned plates consisting of HOM and EG(1,1)-profiled layers with and without elastic foundation against the with (a) =50 and (b) =100
a)
b)
Figure 4. Variation of NLF/LF for single layer and (0°), (0°/90°) and (0°/90°/0°) aligned plates consisting of EG(1,0) and EG(-1,0)-profiled layers with and without elastic foundation against the with (a) =50 and (b) =100
a)
b)
Figure 5. Variation of NLF/LF for single layer and (90°/0°/90°), (0°/90°/90°/0°) and (90°/0°/0°/90°) aligned plates consisting of HOM and EG(1,1)-profiled layers with and without elastic foundation against the with (a) =50 and (b) =100
a)
b)
Figure 6. Variation of NLF/LF for single layer and (90°/0°/90°), (0°/90°/90°/0°) and (90°/0°/0°/90°) aligned plates consisting of EG(1,0) and EG(-1,0)-profiled layers with and without elastic foundation against the with (a) =50 and (b) =100
SUGGESTION 5: Can the current modified Lindstedt-Poincaré method be used to analyze other types of composite structures?
EXPLANATION 5: Thanks. The present modified Lindstedt-Poincaré method can certainly be used to analyze other types of composite structures. For this, it is necessary to transform the nonlinear ordinary differential equation into the standard form we obtained.
Thank you for your constructive and positive report.

Reviewer 3 Report
Comments and Suggestions for Authors
The authors present the solution of nonlinear vibration problem of laminated plates made of exponentially graded orthotropic layers interaction with elastic foundations within framework of Kirchhoff Love theory. Using the Galerkin method authors reduce the problem to non-linear differential equation, which is investigated with help of series in respect to an artificial small parameter. The paper contains sufficient quantity of numerical calculations, including comparison with the known results in literature. On my sight, the work is mainly correct, and the study pretty much is original and novel. However, some misprints and errors can spoil the impression of the paper. I recommend the publication of the manuscript with minor corrections of these points:
1) Authors write in the beginning of paper that “…each EGOL has thickness δ = N/h”. It is mistake, you need δ = h/N”
2) In the sentence after Eq. (19) : “After substituting relations (16) and (17) into Eq. (23),…”. You need to write Eq. (19) instead Eq. (23)
3) Misprint in Eq. (21), two lines under root.
4) Using substitution of Eq. (23), authors transform the differential equation (20) into form (24). Obviously that dt2 = ω2dt2, then in all coefficients (25) must be not ω‑2, but ω2, i.e. α= k1 ω2, β= k2 ω2/ε, γ= k3 ω2/ε.
However if ε is small value, the β and γ are sufficiently large, tends to infinity. I have following question. Will be big values of these coefficients influence on the numerical error of method? Do you investigate numerical stability?
5) Next, it should be noted that the differential equation may be solved exactly in elliptic functions. Moreover there is a case when equation (24) has elementary solution. Try to use the following change of variables
dW/dt = P(W).
On may sight, will be interesting to compare exact nonlinear solution with presented approximation.
Author Response
25.02.2024
EXPLANATION TO REVIEWER 3:
First of all, we would like to thank the highly respected Reviewer 3 for his/her comments and the time spent for them.
Reviewer 3
( ) I would not like to sign my review report
(x) I would like to sign my review report
Quality of English Language
( ) I am not qualified to assess the quality of English in this paper
( ) English very difficult to understand/incomprehensible
( ) Extensive editing of English language required
( ) Moderate editing of English language required
( ) Minor editing of English language required
(x) English language fine. No issues detected
| 
 Yes  | 
 Can be improved  | 
 Must be improved  | 
 Not applicable  | 
|
| 
 Does the introduction provide sufficient background and include all relevant references?  | 
 (x)  | 
 ( )  | 
 ( )  | 
 ( )  | 
| 
 Are all the cited references relevant to the research?  | 
 ( )  | 
 (x)  | 
 ( )  | 
 ( )  | 
| 
 Is the research design appropriate?  | 
 (x)  | 
 ( )  | 
 ( )  | 
 ( )  | 
| 
 Are the methods adequately described?  | 
 ( )  | 
 ( )  | 
 (x)  | 
 ( )  | 
| 
 Are the results clearly presented?  | 
 (x)  | 
 ( )  | 
 ( )  | 
 ( )  | 
| 
 Are the conclusions supported by the results?  | 
 (x)  | 
 ( )  | 
 ( )  | 
 ( )  | 
Comments and Suggestions for Authors
The authors present the solution of nonlinear vibration problem of laminated plates made of exponentially graded orthotropic layers interaction with elastic foundations within framework of Kirchhoff Love theory. Using the Galerkin method authors reduce the problem to non-linear differential equation, which is investigated with help of series in respect to an artificial small parameter. The paper contains sufficient quantity of numerical calculations, including comparison with the known results in literature. On my sight, the work is mainly correct, and the study pretty much is original and novel. However, some misprints and errors can spoil the impression of the paper. I recommend the publication of the manuscript with minor corrections of these points:
SUGGESTION 1: Authors write in the beginning of paper that “…each EGOL has thickness δ = N/h”. It is mistake, you need δ = h/N”
EXPLANATION 1: Thanks. You are right. This typo is corrected in the revised manuscript.
SUGGESTION 2: In the sentence after Eq. (19) : “After substituting relations (16) and (17) into Eq. (23),…”. You need to write Eq. (19) instead Eq. (23)
EXPLANATION 2: Thanks. You are right. This typo is corrected in the revised manuscript.
SUGGESTION 3: Misprint in Eq. (21), two lines under root.
EXPLANATION 3: Thanks. In the revised manuscript, the Eq. (21) is written as follows:
instead of
SUGGESTION 4: Using substitution of Eq. (23), authors transform the differential equation (20) into form (24). Obviously that dt2 = ω2dt2, then in all coefficients (25) must be not ω‑2, but ω2, i.e. α= k1 ω2, β= k2 ω2/ε, γ= k3 ω2/ε.
However if ε is small value, the β and γ are sufficiently large, tends to infinity. I have following question. Will be big values of these coefficients influence on the numerical error of method? Do you investigate numerical stability?
EXPLANATION 4: Thanks. You are right. The typo is made not in the Eq. (24), but in the previous relation (23). Here is a dimensionless parameter. Since the unit of is sec and the unit of frequency is 1/sec, must be equal to . Namely, it must be instead of .This typo is corrected in the revised manuscript. This correction will also clarify the second part of the question.
SUGGESTION 5: Next, it should be noted that the differential equation may be solved exactly in elliptic functions. Moreover there is a case when equation (24) has elementary solution. Try to use the following change of variables dW/dt = P(W).
On may sight, will be interesting to compare exact nonlinear solution with presented approximation.
EXPLANATION 5: Thanks. This is a very interesting suggestion and should be taken into consideration in the future. In this study, since it is aimed to reach the solution with the Lindstedt-Poincaré method and the special case of the non-linear frequency results obtained is in good agreement with the results obtained by the finite element method. Therefore, the topic of the problem and the solution of the problem are provided as a whole.
Thank you for your constructive and positive report.

Reviewer 4 Report
Comments and Suggestions for Authors
Report on The Manuscript ID
mathematics-2889111
February 21, 2024
Dear Editor,
The mathematical modelling of mechanical properties of exponentially graded
composite laminated plates and elastic foundations are presented. The nonlin-
ear dynamic model of exponentially graded composite laminated plates is in-
troduced. The nonlinear di§erential equations for laminated plates consisting
of exponentially graded composite layers interaction with elastic foundations
are solved using the modiÖed Lindstedt-PoincarÈ method. After performing nu-
merical simulations to verify the validity of the analytical expressions, numerical
examples are presented to verify the speciÖcity of NL vibrations of exponentially
graded composite laminated plates.
The structure of the manuscript is clear and the analysis is acceptable.
- Can the new additions to this paper be more clearly identiÖed and con-
trasted with each of the studies stated in the introduction?
- Can the same method be used for other types?
- Rewrite the conclusion section

Report on The Manuscript ID
mathematics-2889111
February 21, 2024
Dear Editor,
The mathematical modelling of mechanical properties of exponentially graded
composite laminated plates and elastic foundations are presented. The nonlin-
ear dynamic model of exponentially graded composite laminated plates is in-
troduced. The nonlinear di§erential equations for laminated plates consisting
of exponentially graded composite layers interaction with elastic foundations
are solved using the modiÖed Lindstedt-PoincarÈ method. After performing nu-
merical simulations to verify the validity of the analytical expressions, numerical
examples are presented to verify the speciÖcity of NL vibrations of exponentially
graded composite laminated plates.
The structure of the manuscript is clear and the analysis is acceptable.
- Can the new additions to this paper be more clearly identiÖed and con-
trasted with each of the studies stated in the introduction?
- Can the same method be used for other types?
- Rewrite the conclusion section
Author Response
25.02.2024
EXPLANATION TO REVIEWER 4:
First of all, we would like to thank the highly respected Reviewer 4 for his/her comments and the time spent for them.
Reviewer 4
( ) I would not like to sign my review report
(x) I would like to sign my review report
Quality of English Language
( ) I am not qualified to assess the quality of English in this paper
( ) English very difficult to understand/incomprehensible
( ) Extensive editing of English language required
( ) Moderate editing of English language required
(x) Minor editing of English language required
( ) English language fine. No issues detected
| 
 Yes  | 
 Can be improved  | 
 Must be improved  | 
 Not applicable  | 
|
| 
 Does the introduction provide sufficient background and include all relevant references?  | 
 (x)  | 
 ( )  | 
 ( )  | 
 ( )  | 
| 
 Are all the cited references relevant to the research?  | 
 (x)  | 
 ( )  | 
 ( )  | 
 ( )  | 
| 
 Is the research design appropriate?  | 
 (x)  | 
 ( )  | 
 ( )  | 
 ( )  | 
| 
 Are the methods adequately described?  | 
 (x)  | 
 ( )  | 
 ( )  | 
 ( )  | 
| 
 Are the results clearly presented?  | 
 (x)  | 
 ( )  | 
 ( )  | 
 ( )  | 
| 
 Are the conclusions supported by the results?  | 
 (x)  | 
 ( )  | 
 ( )  | 
 ( )  | 
Comments and Suggestions for Authors
Dear Editor,
The mathematical modelling of mechanical properties of exponentially graded composite laminated plates and elastic foundations are presented. The nonlinear dynamic model of exponentially graded composite laminated plates is introduced. The nonlinear differential equations for laminated plates consisting of exponentially graded composite layers interaction with elastic foundations are solved using the modifieds Lindstedt-PoincarÈ method. After performing numerical simulations to verify the validity of the analytical expressions, numerical examples are presented to verify the specificity of NL vibrations of exponentially graded composite laminated plates.
The structure of the manuscript is clear, and the analysis is acceptable.
SUGGESTION 1: Can the new additions to this paper be more clearly identified and contrasted with each of the studies stated in the introduction?
EXPLANATION 1: Thanks. The following statements have been added to the introduction of the revised manuscript.
The results of this theoretical study of the nonlinear vibration of laminated plates composed of EGOLs in elastic media have practical implications for optimizing structural design, material selection, and performance in aerospace, marine, automotive, and related fields. The knowledge gained can also contribute to nonlinear vibration control and structural health monitoring. In addition, the Lindstedt-Poincaré method used to solve this problem can be applied to other laminated structural systems…..
The analysis conducted in this study makes an innovative contribution to the existing literature by comprehensively assessing the linear and nonlinear vibration behavior of laminated plates consisting of EGOLs in the continuous elastic medium.
SUGGESTION 2: Can the same method be used for other types?
EXPLANATION 2: Thanks. The present modified Lindstedt-Poincaré method can certainly be used to analyze other types of composite structures. For this, it is necessary to transform the nonlinear ordinary differential equation into the standard form we obtained.
SUGGESTION 3: Rewrite the conclusion section
EXPLANATION 3: Thanks. In the revised manuscript, the conclusion is expanded as follows:
………The current analytical solution may be a valid option for comparison with finite element simulations. In addition, lamination schemes within the scope of two-dimensional theories in continuous environments enable fast and accurate solution of the nonlinear vibration problem of cross-ply plates made of orthotropic materials. Therefore, compared with reliable numerical models, it can be applied to lattice and honeycomb plates without significant computational effort, as well as to plates composed of heterogeneous composite materials.
The numerical results are interpreted and generalized as follows:
- The NL frequency values increase when and ratios increase, in cases with and without ground, in all alignments and layers, and in exponentially graded orthotropic material profiles.
 - In the homogeneous orthotropic and exponentially graded orthotropic material cases, increasing supports the increase of the NL frequency, but also reveals that the ground effect makes the NL frequency values significant enough to be considered.
 - Since each of the EG(1,1), EG(1,0) and EG(-1,0) profiles has their own unique effects on the NL frequency, separate evaluation and interpretation must be made for each profile.
 - NL frequency values of laminated plates consisting of EG(-1.0) profile layers are higher than the values of the homogeneous case, and are lower in other exponentially graded orthotropic material profiles.
 - The effect of exponential graduation profiles on the NL frequency changes when the layer layout or number of layers changes.
 - Taking the ground into account reduces the influence of the layer arrangement on the NL frequency.
 - As the model effect on NL frequency values of laminated plate consisting of layers with exponentially graded orthotropic material profiles is compared among themselves, the most significant effect occurs in the plate with the EG(-1.0)
 - Considering the ground weakens the influence of EG profiles and layer arrangement on the NL frequency.
 - Although the NLF/LF ratio increases when the increases in plates starting from (0°/…) or (90°/…)-array layers, in cases with and without ground, that ratio increases more clearly in some aligned plates in the case with ground.
 - In the case without a ground, the influences of exponentially graded orthotropic material profiles on the NLF/LF ratio are weak at small values of , while those effects increase for all alignments with the subsequent increase of , and show a more significant increase in plates with alignments starting with (0°/…) and are independent of the
 - In the presence of ground effect, when the ratio increases for the selected f, the effect of exponentially graded orthotropic material profiles on the NLF/LF ratio increases significantly depending on the layer arrangement and number and show more significant increase in plates with alignments starting with (90°/…).
 - In the case of without the ground, when increases for the selected , the influence of exponentially graded orthotropic material profiles on the NLF/LF ratio increases, whereas in the presence of the ground, those effects decrease, although they are much more pronounced.
 - When the NLF/LF ratio of exponentially graded orthotropic material profiled plates with different alignments consisting of two, three and four layers on the ground is compared with the single layer plate (0°), the alignment effects become evident with the increase of .
 - In all layered plates, the influence of the soil on the NLF/LF ratio decreases significantly as increases, while the increase of the increases that effect significantly.

Thank you for your constructive and positive report.

Reviewer 5 Report
Comments and Suggestions for Authors
The solution of the nonlinear (NL) vibration problem of laminated
plates made of
exponentially graded orthotropic layers (EGOLs) interaction with elastic
foundations within
Kirchhoff Love Theory (KLT) is developed using the modified
Lindstedt-Poincaré method for the first time. Young’s modulus and material
density of the orthotropic layers of laminated plates are assumed to vary
exponentially in the thickness direction, and Poisson’s ratio is assumed to
be constant.
Although the topic is of a certain interest for a lot of researchers, the paper under consideration does not meet the high standards of "mathematics ".
For the convenience of the authors, some remarks are listed below:
- The Introduction is unsatisfactory: the author should mention the reasons that suggest to consider method and equation.
- What is the new addition to your paper regarding previous studies in 4.5?
- plz add some information about differential equations see, https://doi.org/10.1016/j.jksus.2023.102730
- Throughout the paper there are several typos (unusual expressions, inadequate punctuation, ...). The English of the paper should be deeply revised.
- The proofs of the results do not exhibit particular novelties in terms of techniques.
- Section ("Conclusion") does not add further details. It could be of

Author Response
25.02.2024
EXPLANATION TO REVIEWER 5:
First of all, we would like to thank the highly respected Reviewer 5 for his/her comments and the time spent for them.
Reviewer 5:
(x) I would not like to sign my review report
( ) I would like to sign my review report
Quality of English Language
(x) I am not qualified to assess the quality of English in this paper
( ) English very difficult to understand/incomprehensible
( ) Extensive editing of English language required
( ) Moderate editing of English language required
( ) Minor editing of English language required
( ) English language fine. No issues detected
| 
 Yes  | 
 Can be improved  | 
 Must be improved  | 
 Not applicable  | 
|
| 
 Does the introduction provide sufficient background and include all relevant references?  | 
 ( )  | 
 (x)  | 
 ( )  | 
 ( )  | 
| 
 Are all the cited references relevant to the research?  | 
 ( )  | 
 (x)  | 
 ( )  | 
 ( )  | 
| 
 Is the research design appropriate?  | 
 ( )  | 
 (x)  | 
 ( )  | 
 ( )  | 
| 
 Are the methods adequately described?  | 
 ( )  | 
 (x)  | 
 ( )  | 
 ( )  | 
| 
 Are the results clearly presented?  | 
 ( )  | 
 (x)  | 
 ( )  | 
 ( )  | 
| 
 Are the conclusions supported by the results?  | 
 ( )  | 
 (x)  | 
 ( )  | 
 ( )  | 
Comments and Suggestions for Authors
Dear authors, see the file
Submission Date 08 February 2024
Date of this review 20 Feb 2024 15:22:44
The solution of the nonlinear (NL) vibration problem of laminated plates made of exponentially graded orthotropic layers (EGOLs) interaction with elastic foundations within Kirchhoff Love Theory (KLT) is developed using the modified Lindstedt-Poincaré method for the first time. Young’s modulus and material density of the orthotropic layers of laminated plates are assumed to vary exponentially in the thickness direction, and Poisson’s ratio is assumed to be constant. Although the topic is of a certain interest for a lot of researchers, the paper under consideration does not meet the high standards of "mathematics ".
For the convenience of the authors, some remarks are listed below:
SUGGESTION 1: The Introduction is unsatisfactory: the author should mention the reasons that suggest to consider method and equation.
EXPLANATION 1: Thanks. The following statements have been added to the introduction of the revised manuscript.
The results of this theoretical study of the nonlinear vibration of laminated plates composed of EGOLs in elastic media have practical implications for optimizing structural design, material selection, and performance in aerospace, marine, automotive, and related fields. The knowledge gained can also contribute to nonlinear vibration control and structural health monitoring. In addition, the Lindstedt-Poincaré method used to solve this problem can be applied to other laminated structural systems…..
The analysis conducted in this study makes an innovative contribution to the existing literature by comprehensively assessing the linear and nonlinear vibration behavior of laminated plates consisting of EGOLs in the continuous elastic medium.
SUGGESTION 2: What is the new addition to your paper regarding previous studies in 4.5?
EXPLANATION 2: The original parts of the manuscript are presented in the modelling, derivation and solution of basic relations and equations of laminated plates consisting of exponentially graded orthotropic layers, and numerical analysis sections.
SUGGESTION 3: plz add some information about differential equations see, https://doi.org/10.1016/j.jksus.2023.102730
EXPLANATION 3: Thanks. The following work is used in the revised manuscript because it is related to our study:
Althubiti, S. Nonlinear third-order differential equations with distributed delay: Some new oscillatory solutions. J King Saud Uni. – Sci. 2023, 35(6), 102730
SUGGESTION 4: Throughout the paper there are several typos (unusual expressions, inadequate punctuation, ...). The English of the paper should be deeply revised.
EXPLANATION 4: Thanks. We have made a careful check throughout the whole manuscript, and tried to identify and eliminate all the grammatical mistakes.
SUGGESTION 5: The proofs of the results do not exhibit particular novelties in terms of techniques.
EXPLANATION 5: Thanks. The main purpose of the proofs is to show that the obtained formulas are in agreement with the results obtained by finite elements and other numerical and analytical methods. The original parts of the manuscript are presented in the modelling, derivation and solution of basic relations and equations, and numerical analysis sections.
SUGGESTION 6: Section ("Conclusion") does not add further details. It could be of interest a description of some open problems related to the equation under consideration.
EXPLANATION 6: Thanks. You are right. The following statements have been added to the introduction of the revised manuscript.
This study is devoted to solving the problem of NL vibration of laminated plates consisting of EGOLs interacting with two-parameter elastic foundation using the Lindstedt-Poincaré method. The NL fundamental relations and basic equations of laminated plates with EGOM profiles in an elastic medium are derived. Amplitude-dependent expressions for the NL frequency are obtained using Galerkin and the modified Lindstedt-Poincaré method. The accuracy of the proposed solution is confirmed by comparing it with reliable results in the literature. The current analytical solution may be a valid option for comparison with finite element simulations. In addition, lamination schemes within the scope of two-dimensional theories in continuous environments enable fast and accurate solution of the nonlinear vibration problem of cross-ply plates made of orthotropic materials. Therefore, compared with reliable numerical models, it can be applied to lattice and honeycomb plates without significant computational effort, as well as to plates composed of heterogeneous composite materials. The numerical results are interpreted and generalized as follows:
- The NL frequency values increase when and ratios increase, in cases with and without ground, in all alignments and layers, and in exponentially graded orthotropic material profiles.
 - In the homogeneous orthotropic and exponentially graded orthotropic material cases, increasing supports the increase of the NL frequency, but also reveals that the ground effect makes the NL frequency values significant enough to be considered.
 - Since each of the EG(1,1), EG(1,0) and EG(-1,0) profiles has their own unique effects on the NL frequency, separate evaluation and interpretation must be made for each profile.
 - NL frequency values of laminated plates consisting of EG(-1.0) profile layers are higher than the values of the homogeneous case, and are lower in other exponentially graded orthotropic material profiles.
 - The effect of exponential graduation profiles on the NL frequency changes when the layer layout or number of layers changes.
 - Taking the ground into account reduces the influence of the layer arrangement on the NL frequency.
 - As the model effect on NL frequency values of laminated plate consisting of layers with exponentially graded orthotropic material profiles is compared among themselves, the most significant effect occurs in the plate with the EG(-1.0)
 - Considering the ground weakens the influence of EG profiles and layer arrangement on the NL frequency.
 - Although the NLF/LF ratio increases when the increases in plates starting from (0°/…) or (90°/…)-array layers, in cases with and without ground, that ratio increases more clearly in some aligned plates in the case with ground.
 - In the case without a ground, the influences of exponentially graded orthotropic material profiles on the NLF/LF ratio are weak at small values of , while those effects increase for all alignments with the subsequent increase of , and show a more significant increase in plates with alignments starting with (0°/…) and are independent of the
 - In the presence of ground effect, when the ratio increases for the selected f, the effect of exponentially graded orthotropic material profiles on the NLF/LF ratio increases significantly depending on the layer arrangement and number and show more significant increase in plates with alignments starting with (90°/…).
 - In the case of without the ground, when increases for the selected , the influence of exponentially graded orthotropic material profiles on the NLF/LF ratio increases, whereas in the presence of the ground, those effects decrease, although they are much more pronounced.
 - When the NLF/LF ratio of exponentially graded orthotropic material profiled plates with different alignments consisting of two, three and four layers on the ground is compared with the single layer plate (0°), the alignment effects become evident with the increase of .
 - In all layered plates, the influence of the soil on the NLF/LF ratio decreases significantly as increases, while the increase of the increases that effect significantly.

After addressing these minor revisions in accordance with the suggestions above, the paper should be considered for acceptance.
Thank you for your constructive and positive report.